# Complementary shifts in photoreceptor spectral tuning unlock the full adaptive potential of ultraviolet vision in birds

Matthew B Toomey[1], Olle Lind[2], Rikard Frederiksen[3], Robert W Curley Jr[4], Ken M Riedl[5,6], David Wilby[7], Steven J Schwartz[5], Christopher C Witt[8,9], Earl H Harrison[10], Nicholas W Roberts[7], Misha Vorobyev[11], Kevin J McGraw[12], M Carter Cornwall[3], Almut Kelber[13], Joseph C Corbo[1]*

[1]Department of Pathology and Immunology, Washington University School of Medicine, St. Louis, United States; [2]Department of Philosophy, Lund University, Lund, Sweden; [3]Department of Physiology and Biophysics, Boston University, Boston, United States; [4]College of Pharmacy, The Ohio State University, Columbus, United States; [5]Department of Food Science and Technology, The Ohio State University, Columbus, United States; [6]Nutrient and Phytochemical Shared Resource of the OSU-Comprehensive Cancer Center, Columbus, United States; [7]School of Biological Sciences, University of Bristol, Bristol, United Kingdom; [8]Department of Biology, University of New Mexico, Albuquerque, United States; [9]Museum of Southwestern Biology, University of New Mexico, Albuquerque, United States; [10]Department of Human Nutrition, The Ohio State University, Columbus, United States; [11]Department of Optometry and Vision Science, University of Auckland, Auckland, New Zealand; [12]School of Life Sciences, Arizona State University, Tempe, United States; [13]Department of Biology, Lund University, Lund, Sweden

*For correspondence: jcorbo@wustl.edu

Competing interests: The authors declare that no competing interests exist.

**Abstract** Color vision in birds is mediated by four types of cone photoreceptors whose maximal sensitivities ($\lambda_{max}$) are evenly spaced across the light spectrum. In the course of avian evolution, the $\lambda_{max}$ of the most shortwave-sensitive cone, SWS1, has switched between violet ($\lambda_{max} > 400$ nm) and ultraviolet ($\lambda_{max} < 380$ nm) multiple times. This shift of the SWS1 opsin is accompanied by a corresponding short-wavelength shift in the spectrally adjacent SWS2 cone. Here, we show that SWS2 cone spectral tuning is mediated by modulating the ratio of two apocarotenoids, galloxanthin and 11′,12′-dihydrogalloxanthin, which act as intracellular spectral filters in this cell type. We propose an enzymatic pathway that mediates the differential production of these apocarotenoids in the avian retina, and we use color vision modeling to demonstrate how correlated evolution of spectral tuning is necessary to achieve even sampling of the light spectrum and thereby maintain near-optimal color discrimination.

## Introduction

Color vision systems are based on visual pigments consisting of a retinoid chromophore bound to an opsin protein (*Yokoyama, 2000*). When the chromophore absorbs light, it changes conformation and initiates a signaling cascade. The peak spectral sensitivity of the visual pigment is determined by the amino acid composition of the opsin and the structure of the chromophore (*Yokoyama, 2000; Wang et al., 2014; Bridges, 1972*). Color vision is achieved by comparing the stimulation of receptors that are maximally sensitive to different wavelengths of light (*Goldsmith, 1990*), but overlap of

**eLife digest** The pioneering eye doctor André Rochon-Duvigneaud once wrote that "a bird is a wing guided by an eye". With this statement, he underscored the sophistication of the bird's eye, which surpasses our own in several respects. Compared to humans who have three types of cone photoreceptor, birds have four, meaning they can see an extra dimension of color. Birds precisely tune their violet-, blue-, green- and red-sensitive cones by coupling light-sensitive proteins with light-filtering pigments called carotenoids. This combination of sensors and filters increases the number of colors a bird can see.

Another exceptional aspect of bird vision is that some species – for example finches and sparrows – can see ultraviolet (UV) light. This ability results from a change in the light-sensitive protein within the violet cone photoreceptor that shifts its sensitivity towards UV light. This expansion of vision into the UV is complemented by a shift in the sensitivity of the blue cone photoreceptor. However, it is not well understood exactly how the sensitivity of the blue cone is shifted and how this shift impacts color vision.

To find answers to these questions, Toomey et al. characterized the light-filtering carotenoid pigments from bird species with violet or UV sensitivity, and used computational models of bird vision to predict how these pigments affect the number of colors a bird can see. This approach revealed that blue cone sensitivity is fine-tuned through a change in the chemical structure of the light-filtering carotenoid pigments within the photoreceptor. Computational models also indicated the sensitivity of the violet and blue cones must shift in a coordinated manner to maximize the number of colors a bird can see.

These results suggest that both blue and violet cone cells have been fine-tuned during evolution to enhance color vision in birds. An important next step is to investigate the underlying molecular mechanisms that coordinate the modification of the carotenoid pigments and the tuning of light-sensitive proteins in a wide range of bird species.

the receptors' sensitivity spectra limits color discrimination (*Govardovskii et al., 2000*; *Dyer, 1999*; *Vorobyev and Osorio, 1998*; *Vorobyev, 2003*). Birds have overcome this limitation by narrowing the spectral bandwidth of some of their photoreceptors through the use of a specialized optical organelle, the pigmented cone oil droplet (*Figure 1a–b*). These oil droplets are located in the path of light through the receptor and act as long-pass cutoff filters matched to the visual pigment sensitivity of each cone subtype. Light filtering by the droplet narrows the spectral bandwidth of receptor sensitivity, thereby reducing sensitivity overlap between spectrally adjacent cone types (*Figure 1c–d*). Theoretical considerations suggest that this adaptation enhances color discrimination and color constancy (*Vorobyev, 2003*; *Vorobyev et al., 1998*).

All birds can be classified into two groups: violet-sensitive species (VS), whose most short-wavelength sensitive cone (SWS1) is maximally sensitive to violet light ($\lambda_{max} > 400$ nm), and ultraviolet-sensitive species (UVS), whose SWS1 cone is maximally sensitive to ultraviolet light ($\lambda_{max} < 380$ nm) (*Cuthill et al., 2000*; *Ödeen and Håstad, 2003*; *2013*; *Shi et al., 2001*). In the course of avian evolution, the switch between violet and ultraviolet sensitivity has occurred at least 14 times (*Ödeen and Håstad, 2013*). Although the selection pressures driving this switch are not well understood, the lability of this change likely reflects its relatively simple molecular basis. The essence of this switch is a short-wavelength shift in the peak sensitivity of the SWS1 opsin that can be mediated by as little as a single amino acid substitution (e.g. S90C) in the second transmembrane helix (*Shi et al., 2001*; *Altun et al., 2011*; *Hauser et al., 2014*). This short-wavelength shift of the SWS1 cone expands the spectral sensitivity of the visual system and is predicted to enhance color discrimination (*Vorobyev et al., 1998*; *Cuthill et al., 2000*). However, a shift in the SWS1 cone alone disrupts the even spacing of receptor sensitivities across the visible spectrum and necessitates corresponding shifts in the sensitivity of the other single cone photoreceptors to maintain even spacing.

Accordingly, between VS and UVS species, the sensitivity of the short-wavelength sensitive 2 (SWS2) cone is tuned to maximize its spectral separation from the adjacent SWS1 and medium-

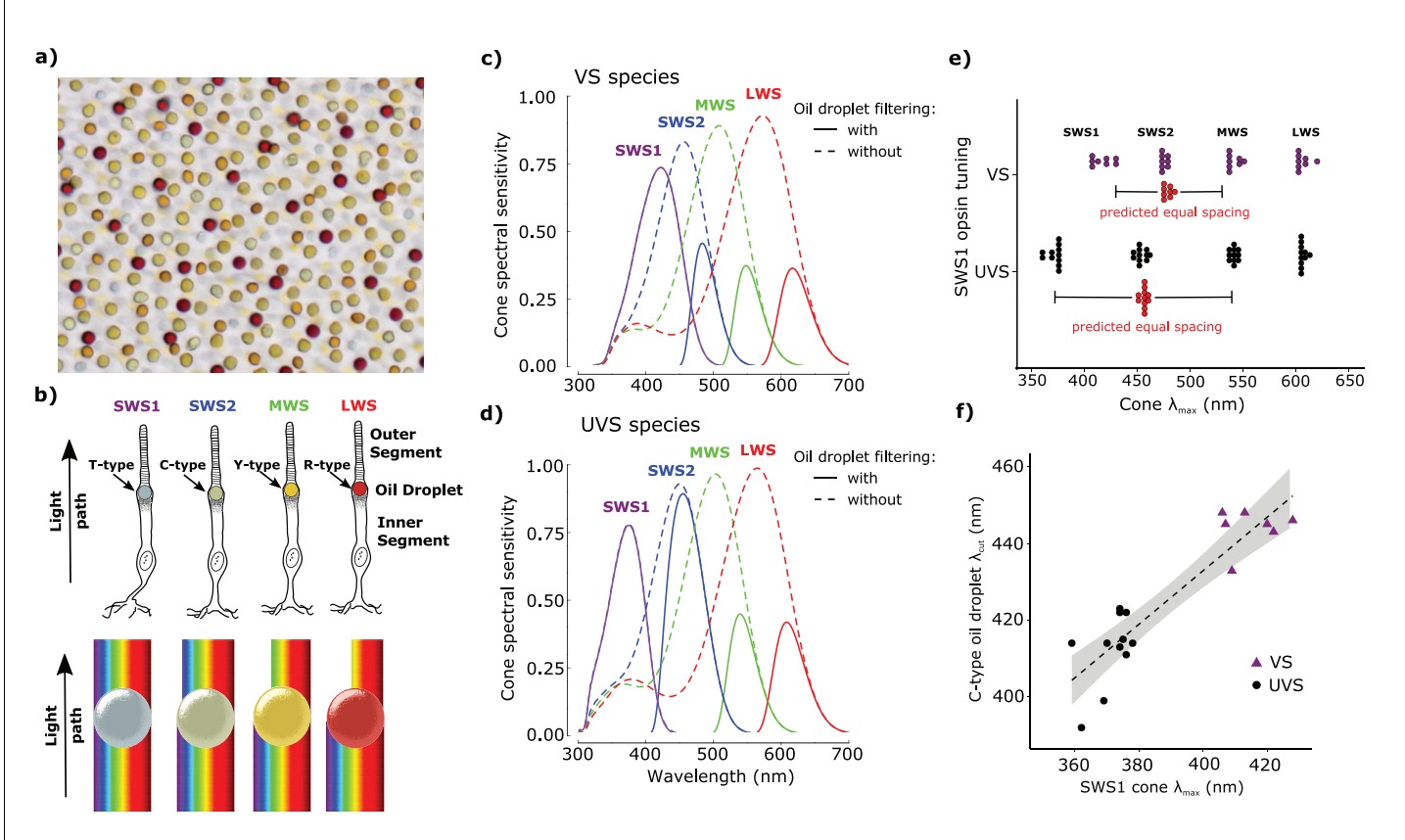

**Figure 1.** Avian color vision is mediated by four single cone photoreceptors that are tuned by cone oil droplet spectral filters. (**a**) A flat-mounted chicken retina under brightfield illumination that shows the distinctive pigmentation of the cone oil droplets. (**b**) A diagram of the avian single cone photoreceptors showing the relative position of the oil droplet within the cells (top) and a representation of the spectral filtering cutoff effects of the droplet (bottom). The spectral sensitivity of the single cone photoreceptors of the (**c**) chicken (VS species) and (**d**) zebra finch (UVS species) with (solid lines) or without (broken lines) the filtering of cone oil droplets. These spectra are scaled to reflect the decrement in absolute sensitivity resulting from oil droplet and ocular media spectral filtering. The SWS1 cone contains a transparent oil droplet, and its short-wavelength sensitivity is limited only by the filtering of the ocular media. (**e**) The peak sensitivity ($\lambda_{max}$) of the four single cone photoreceptors of VS (purple) and UVS (black) species calculated from microspectrophotometric measurements of visual pigment sensitivity and oil droplet filtering in published reports (**Supplementary file 2**). The red points shown below the SWS2 $\lambda_{max}$ values are the $\lambda_{max}$ for each species that would maximize spectral separation from adjacent receptors ($\lambda_{max\ of\ maximum\ separation} = \lambda_{max\ SWS1} + (\lambda_{max\ MWS} - \lambda_{max\ SWS1})/2$). The actual SWS2 $\lambda_{max}$ of the UVS species do not differ significantly from these predicted values (paired t-test, t = − 1.27, p = 0.23) and the $\lambda_{max}$ of the VS species are an average of only 3.6 nm shorter than the predicted values (paired t-test, t = −2.56, p = 0.04). (**f**) Across bird species there is a significant correlation between the spectral tuning of the SWS1 visual pigment and the blue cone oil droplet filtering cutoff (phylogenetic generalized linear model: t = 5.55, p<0.0001, $r^2$ = 0.63). Each point represents a different species; the dashed line is a linear regression through the points with 95% confidence interval shown in gray.

wavelength sensitive (MWS) cones (**Figure 1c–e**). However, in contrast to the opsin-mediated shift of the SWS1 cone sensitivity, the spectral shift of the SWS2 cone is driven primarily by changes in the filtering of the C-type oil droplet (**Hart and Vorobyev, 2005**). Among bird species, there is a strong positive correlation between the spectral sensitivity of the SWS1 opsin and the spectral cut-off wavelength of oil droplet filtering in the SWS2 cone (**Figure 1f**) (**Hart and Vorobyev, 2005**). This finding suggests that distinct mechanisms of spectral tuning are coordinated to ensure even sampling of the visible spectrum.

To elucidate the mechanism underlying these shifts in oil droplet spectral filtering, knowledge of the pigments that mediate filtering is required. It has long been known that cone oil droplets are pigmented with carotenoids, a class of terpenoid molecules whose light absorption properties are determined by their degree of π-electron conjugation (**Capranica, 1877**; **Goldsmith et al., 1984**; **Chábera et al., 2009**). Birds cannot synthesize carotenoids *de novo* and must acquire them through

their diet (**McGraw, 2006**). However, birds can metabolize these diet-derived carotenoids to change their degree of conjugation and thereby shift their absorption spectrum toward longer or shorter wavelengths (**Schiedt, 1998**; **Bhosale et al., 2007**).

The C-type oil droplets within the SWS2 cone of birds is pigmented with apocarotenoids, which are believed to be products of the asymmetrical oxidative cleavage of dietary precursor carotenoids (**Goldsmith et al., 1984**; **Toomey et al., 2015**). These molecules have relatively short systems of conjugation and specifically absorb short-wavelength light (**Goldsmith et al., 1984**; **Toomey et al., 2015**). In the violet-sensitive chicken (*Gallus gallus*), the C-type droplet contains primarily galloxanthin with a peak absorbance at 402 nm as well as an apocarotenoid of undetermined structure (Apo2) with short-wavelength-shifted peak absorbance at 380 nm (**Figure 2a**) (**Toomey et al., 2015**). Here, we hypothesized that the shift in the spectral tuning of the SWS2 cone between VS and UVS species is mediated by the differential accumulation of these two apocarotenoids in the C-type cone oil droplet. To test this hypothesis, we biochemically characterized the apocarotenoids in the retinas and cone oil droplets of VS and UVS species. We then investigated the metabolism of these apocarotenoid pigments and identified a putative enzymatic pathway for their production in the eye. We characterized the enzymatic activity and expression pattern of candidate components of this pathway. Finally, we used computational models of avian visual sensitivity and color discrimination to test the hypothesis that complementary shifts in C-type oil droplet filtering are necessary to realize the full functional potential of the VS to UVS switch of the SWS1 opsin.

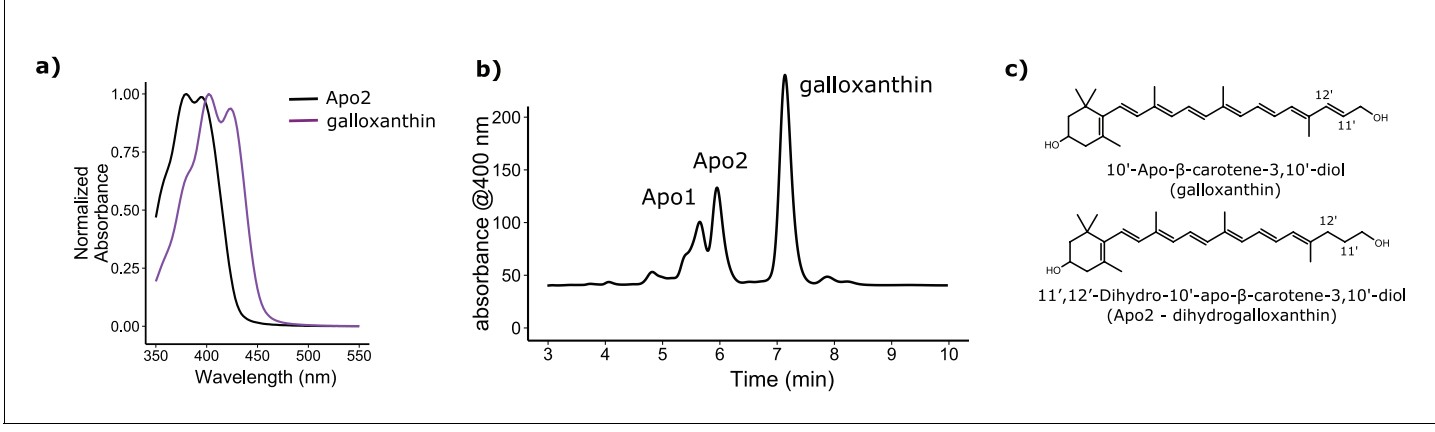

**Figure 2.** The avian retina contains multiple apocarotenoids that absorb different portions of the light spectrum. (**a**) The absorbance spectrum of the major apocarotenoid pigments in the chicken C-type oil droplets. (**b**) A representative HPLC chromatogram of the apocarotenoids in whole retina extracts of the chicken retina. Apo1 has an absorbance spectrum similar to galloxanthin with more pronounced fine structure, suggesting an ε-ring configuration (**Figure 2—figure supplement 2**). This pigment is not a major component of the C-type oil droplets (**Toomey et al., 2015**). (**c**) The proposed chemical structure of dihydrogalloxanthin and known structure of galloxanthin with the positon of the 11′,12′ double bond indicated.

The following figure supplements are available for figure 2:

**Figure supplement 1.** The absorbance spectra of Apo2 and 12′-Apo-β-carotene-3,12′-diol (fringillixanthin) are nearly identical.

**Figure supplement 2.** The three major apocarotenoids present in the avian retina have distinct light absorbance spectra.

**Figure supplement 3.** The structure and predicted wavelength of maximum absorbance for galloxanthin and the eight possible monosaturated forms of galloxanthin.

**Figure supplement 4.** Confirmation of the 11′,12′ saturation of dihydrogalloxanthin via derivatization and ozonolysis.

# Results

## Determination of the molecular structure of the retinal apocarotenoid, Apo2

To elucidate the mechanism whereby apocarotenoids tune the spectral filtering of the C-type oil droplet, we first set out to determine the structure of Apo2. The absorbance spectrum of Apo2 is short-wavelength shifted by 22 nm relative to galloxanthin, suggesting that Apo2 has fewer conjugated double bonds than galloxanthin (*Figure 2a*). Apo2 had previously been observed in the retinas of some bird species and was hypothesized to be a truncated (25 carbons) form of galloxanthin (27 carbons) with seven double bonds in conjugation (*Goldsmith et al., 1984*). To test this hypothesis, we measured the absorbance spectrum of a commercial standard of this 25-carbon apocarotenoid (referred to in the original study as fringillixanthin [*Goldsmith et al., 1984*]) and found that it matched the spectrum of Apo2 almost exactly (*Figure 2—figure supplement 1*). This result supports the hypothesis that Apo2 is fringillixanthin.

However, when we examined the apocarotenoid profile of the chicken retina by liquid chromatography-mass spectral (LC-MS) analysis, we found that the molecular weight of Apo2 was two daltons greater than that of galloxanthin (*Figure 2b*; *Table 1*). This finding suggested that Apo2 has the same number of carbons as galloxanthin but has undergone saturation of one of its double bonds.

To predict the site of this saturation, we calculated the wavelength of maximum absorbance for all eight possible monosaturated forms of galloxanthin based on the Woodward-Fieser and Fieser-Kuhn rules (*Silverstein et al., 1991*), and compared these predictions to the absorbance maximum of Apo2. We found that saturation of the 11',12' double bond of galloxanthin is predicted to cause a 22.5 nm blue-shift in the wavelength of peak absorbance, a close match to the measured shift of 22 nm observed between Apo2 and galloxanthin (*Figure 2—figure supplement 3*).

To determine experimentally if Apo2 is the 11',12'-saturated form of galloxanthin, we derivatized Apo2 with para-nitrobenzoyl chloride, cleaved all non-aromatic double bonds with ozone, and measured the MS/MS spectra of the remaining fragments. Ozonolysis is expected to generate a diagnostic 251 dalton product if Apo2 is saturated at the 11',12' position. To test this prediction, we synthesized a standard for the 251 dalton product, confirmed its structure with [1]H NMR, and then used MS/MS spectroscopy to compare this standard to the products of ozonolysis of derivatized Apo2. The critical product of Apo2 derivatization-ozonolysis had a parent ion and fragmentation pattern identical to that of the standard, confirming the presence of saturation at the 11',12' double bond (*Figure 2—figure supplement 4*). These results indicate that Apo2 is 11',12'-dihydro-10'-Apo-β-carotene-3,10'-diol, which we refer to as dihydrogalloxanthin (*Figure 2c*). This apocarotenoid was previously observed in the skin of black bass (*Micropterus salmoides*); however, its functional role in that tissue is unknown (*Yamashita et al., 1996*; *Yamashita and Matsuno, 1992*).

## C-type oil droplet spectral filtering is determined by the accumulation of two apocarotenoids, galloxanthin and dihydrogalloxanthin

Having established the structure of dihydrogalloxanthin, we next sought to determine its role in oil droplet spectral filtering. We first evaluated the relative contributions of galloxanthin and dihydrogalloxanthin to the apocarotenoid composition of the whole retinas from a range of bird species. We examined the retinal apocarotenoid composition of 21 VS and 24 UVS species across eight avian orders using high performance liquid chromatography (HPLC) (*Supplementary file 1*). We found that the apocarotenoid ratio (dihydrogalloxanthin:galloxanthin) differed significantly between VS and

**Table 1.** The exact mass measurements of major apocarotenoids in the chicken retina. Galloxanthin lost a water molecule in the ionization process resulting in a difference of 18 units in the measurement of mass.

| Carotenoid | m/z observed | m/z theoretical | ppm error | Ion | m/z of intact molecule |
|---|---|---|---|---|---|
| Apo2 | 397.308 | 397.3101 | -5.29 | M+H | 397.308 |
| galloxanthin | 377.2836 | 377.2844 | -2.12 | M+H-H$_2$O | 395.2836 |

UVS species ($F_{1,43}$ = 22.32, p=2.48 x $10^{-5}$, *Figure 3a*). UVS species tended to have higher levels of the short wavelength-shifted dihydrogalloxanthin in their retinas (dihydrogalloxanthin:galloxanthin ratio 0.81 ± 0.09, mean ± SD; n = 21) compared to VS species (0.52 ± 0.27, n = 24, *Figure 3a*). However, this ratio was subject to strong phylogenetic inertia (Blomberg's K = 1.58, Pagel's λ = 0.93, *Figure 3—figure supplement 1*). This means that closely related species tend to have similar apocarotenoid compositions in their retinas and we must account for these correlations to properly assess the relationship between apocarotenoid composition and SWS1 opsin tuning. We therefore repeated the analysis to include phylogeny as part of the model and found that SWS1 opsin tuning remained a significant predictor of dihydrogalloxanthin:galloxanthin ratio (phylogenetic generalized linear model: p=0.012). Overall, these findings suggest that the evolution of the UVS opsin is associated with a relative increase in the concentrations of dihydrogalloxanthin compared to galloxanthin in the retina.

Whole retina measurements, however, may not be the best indicator of the specific composition of the C-type oil droplets. The relationship between whole retina apocarotenoid levels and the

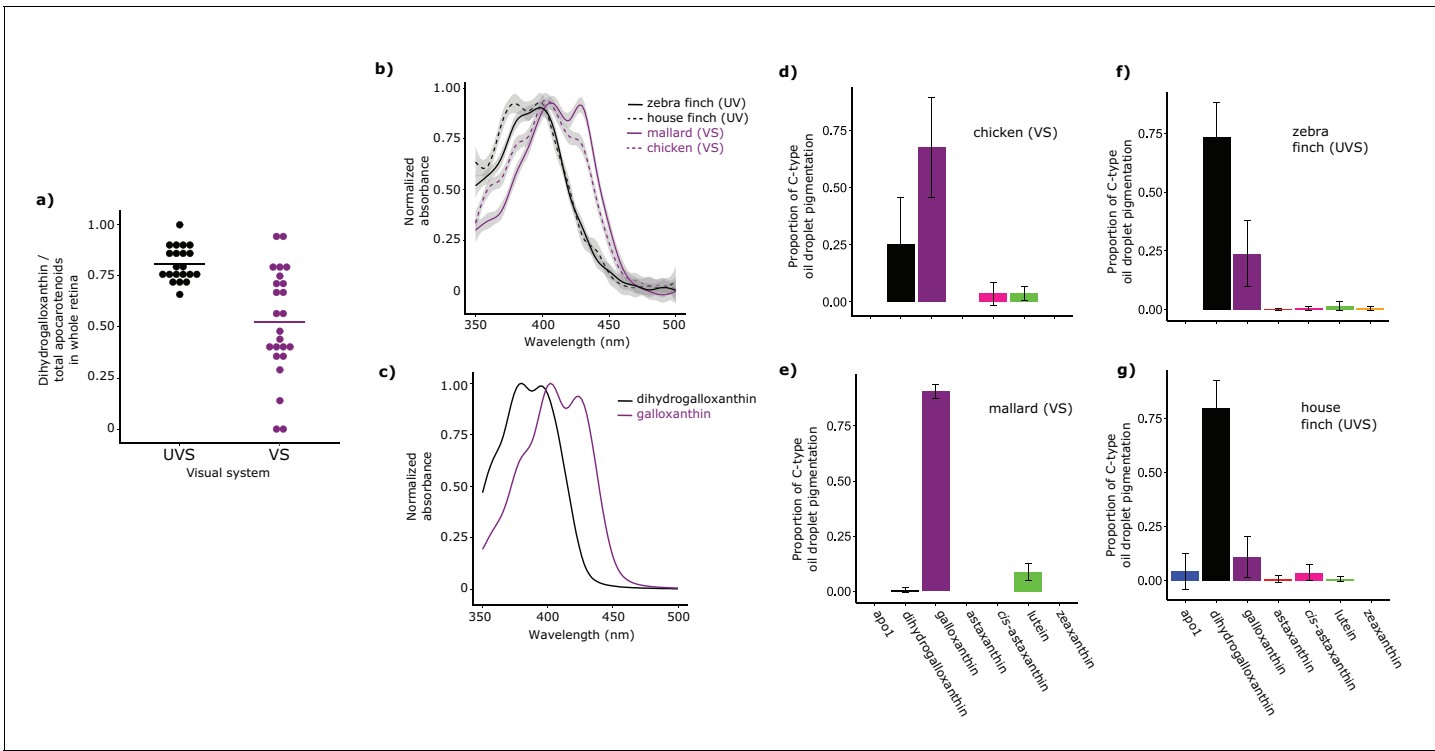

**Figure 3.** Shifts in the spectral filtering of the SWS2 cone between VS and UVS species are the result of changes in apocarotenoid composition. (a) The ratio of dihydrogalloxanthin to total apocarotenoid content in the whole retina differs between UV and VS species. Each point represents a different species, and the bar indicates the mean for each visual system. (b) The UV-Vis absorbance spectrum of the expanded C-type oil droplets of selected UVS and VS species, and (c) the spectrum of galloxanthin and dihydrogalloxanthin for comparison. (d–g) The mean ± S.D. proportion of each carotenoid in the additive mixture of pure spectra that best fits the C-type droplet spectra from (d) chicken (n = 5, VS), (e) mallard (n = 3, VS), (f) zebra finch (n = 5,UVS), and (g) house finch (n = 4, UVS). The pure carotenoid spectra, the observed C-type droplet spectra, and the fitted spectra of each droplet used to estimate oil droplet composition are presented in *Figure 3—figure supplement 2*.

The following source data and figure supplements are available for figure 3:

**Source data 1.** The measured oil droplet spectra, pure carotenoid spectra, and model fit parameters for each measured C-type droplet.

**Figure supplement 1.** The evolution of the UV sensitive SWS1 opsin is associated with changes in whole retina apocarotenoid composition.

**Figure supplement 2.** The observed absorbance spectra of the expanded C-type oil droplets and corresponding fitted spectra produced from additive mixtures of pure carotenoid spectra.

filtering of the C-type droplet is potentially confounded by the additional presence of apocarotenoids in other oil droplet types (*Toomey et al., 2015*). In other photoreceptor types, apocarotenoids accumulate alongside long-wavelength absorbing carotenoids but do not have the same spectral tuning role that they have in SWS2 cones (*Toomey et al., 2015*). Therefore, the precise mixture of apocarotenoids within these other cone types may differ from the ratios in the C-type droplet.

To measure the apocarotenoid content of the C-type droplets directly, we made detailed microspectrophotometric measurements of expanded droplets from two UVS and two VS species (*Figure 3b,c*). We found that the droplet spectra were not well matched by the spectrum of any individual apocarotenoid or carotenoid, suggesting that they consisted of complex mixtures of carotenoids (*Figure 3—figure supplement 2a*). To deconvolve the individual droplet spectra and thereby estimate the exact carotenoid composition of the droplets, we used a linear additive model to find the combination of seven different apocarotenoid/carotenoid spectra that best fit the measured oil droplet spectra. These seven carotenoids used for modeling are the most abundant carotenoids found in the chicken retina (*Toomey et al., 2015*). We previously showed that this method provides accurate estimates of the relative concentrations of carotenoids contained within isolated avian oil droplets (*Toomey et al., 2015*). Using this approach, we were able fit the observed spectra of the C-type oil droplets of each species and explain >98% of the variation (*Figure 3—figure supplement 2b–e*). The fitted spectra consisted primarily of contributions from galloxanthin and dihydrogalloxanthin, with other carotenoid spectra each contributing 9% or less to the best fit. Consistent with our hypothesis, this analysis indicated that the C-type droplets of UVS species contain primarily dihydrogalloxanthin, while the droplets of the VS species contain primarily galloxanthin (*Figure 3d–g*). Thus, birds appear to modulate the relative concentrations of galloxanthin and dihydrogalloxanthin in the C-type droplet to shift the spectral filtering of the SWS2 cone, thereby complementing the spectral tuning of the SWS1 cone. Modulation of the dihydrogalloxanthin: ratio within the SWS2 cone is likely to require controlled synthesis of these molecules from yolk or dietary precursor carotenoids (*Schiedt, 1998*; *Bhosale et al., 2007*). We therefore next sought to determine a potential enzymatic pathway for apocarotenoid production in the retina.

## A proposed enzymatic pathway for apocarotenoid metabolism in the SWS2 cone

Birds acquire carotenoids through their diets and can metabolize these precursors in a variety of ways (*McGraw, 2006*). Radioactive tracer studies indicate that zeaxanthin is the major dietary precursor for the production of galloxanthin in the avian retina (*Schiedt, 1998*; *Bhosale et al., 2007*), but the enzyme(s) mediating this transformation are currently unknown. We hypothesized that the production of dihydrogalloxanthin involves a three-step metabolic pathway in which zeaxanthin is first cleaved at the 9',10' double bond to generate 3-Hydroxy-10'-apo-β-caroten-10'-al; then the aldehyde group is reduced to produce galloxanthin; and finally the 11',12' double bond of galloxanthin is saturated to produce dihydrogalloxanthin (*Figure 4a*).

To search for enzymes mediating these reactions, we examined the transcriptome profiles of avian cone photoreceptors available in a recent report (*Enright et al., 2015a*). In this study, it was reported that a subpopulation of photoreceptors that included the double cones was significantly enriched for β-carotene oxygenase 2 (BCO2), a known carotenoid cleavage enzyme (*Enright et al., 2015a*; *Kiefer et al., 2001*; *Mein et al., 2010*). Like the SWS2 cone, the double cone contains an oil droplet pigmented with galloxanthin, thus we hypothesized that BCO2 might mediate the initial cleavage of zeaxanthin to 3-Hydroxy-10'-apo-β-caroten-10'-al in the synthesis of galloxanthin. To test this hypothesis, we cloned chicken *BCO2* and expressed it in HEK293 cells. We then delivered a zeaxanthin substrate to the cells and characterized the resulting apocarotenoid products by HPLC. Consistent with our hypothesis, cells expressing BCO2 yielded a product consistent with , with a characteristic single absorbance peak at 449 nm (*Figure 4d–e*).

The reduction of 3-Hydroxy-10'-apo-β-caroten-10'-al to galloxanthin is analogous to the retinol dehydrogenase-mediated reduction of all-*trans* retinal to all-*trans* retinol, which is an essential step in the visual cycle (*Parker and Crouch, 2010*). Further examination of the expression profiles of the LWS opsin-expressing photoreceptors revealed that retinol dehydrogenase 12 (RDH12) is the most highly expressed member of the RDH family (*Enright et al., 2015a*). RDH12 is known to act upon a variety of retinal isomers (*Belyaeva et al., 2005*). Thus, we reasoned that this enzyme might be capable of reducing 3-Hydroxy-10'-apo-β-caroten-10'-al to galloxanthin. To test this hypothesis, we

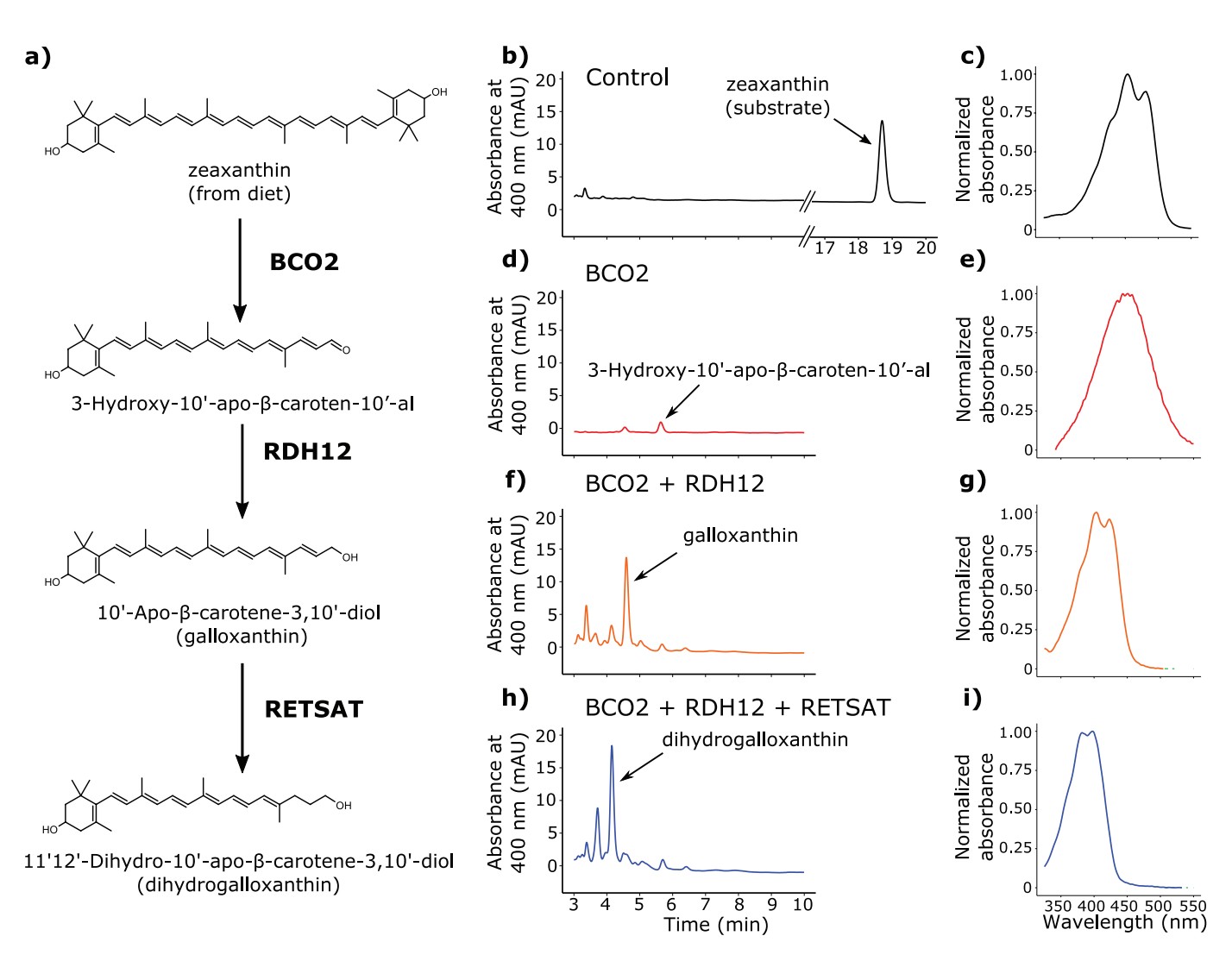

**Figure 4.** BCO2, RDH12, and RETSAT are sufficient to produce dihydrogalloxanthin from zeaxanthin. (a) The proposed metabolic pathway for the production of galloxanthin and dihydrogalloxanthin from dietary zeaxanthin. (b–i) Representative chromatograms and UV-Vis absorbance spectra of apocarotenoid products produced by HEK293 cells expressing a control vector, avian BCO2, RDH12, and/or RETSAT enzymes, and supplemented with zeaxanthin.

cloned chicken *RDH12* and co-expressed it with BCO2 in HEK293 cells. We then delivered a zeaxanthin substrate to the cells and characterized the resulting apocarotenoid products with HPLC. Consistent with our hypothesis, the co-expression of BCO2 and RDH12 yielded galloxanthin (*Figure 4f–g*).

Finally, we observed that the 11',12' saturation of galloxanthin that yields dihydrogalloxanthin is analogous to the 13,14 saturation of retinol that is carried out by retinol saturase (RETSAT) (*Moise et al., 2004*; *2007*). We hypothesized that if RETSAT accepts a range of substrates, it could mediate the formation of dihydrogalloxanthin. To test this hypothesis, we cloned *RETSAT* from the UVS zebra finch and co-expressed it along with BCO2 and RDH12 in HEK293 cells. Supplementation of these cells with zeaxanthin yielded dihydrogalloxanthin, confirming that BCO2, RDH12, and RETSAT are sufficient to produce dihydrogalloxanthin when acting in combination (*Figure 4h–i*).

To confirm that the enzymes in the putative apocarotenoid biosynthetic pathway are expressed in the avian retina, we carried out in situ hybridization on developing chicken retinas. We found that

transcripts for all three enzymes are expressed in the photoreceptor layer of the retina at the time of hatching (*Figure 5a*). We also examined the time course of apocarotenoid accumulation and enzyme transcript levels in the developing chicken retina by HPLC and qPCR, respectively. We first detected

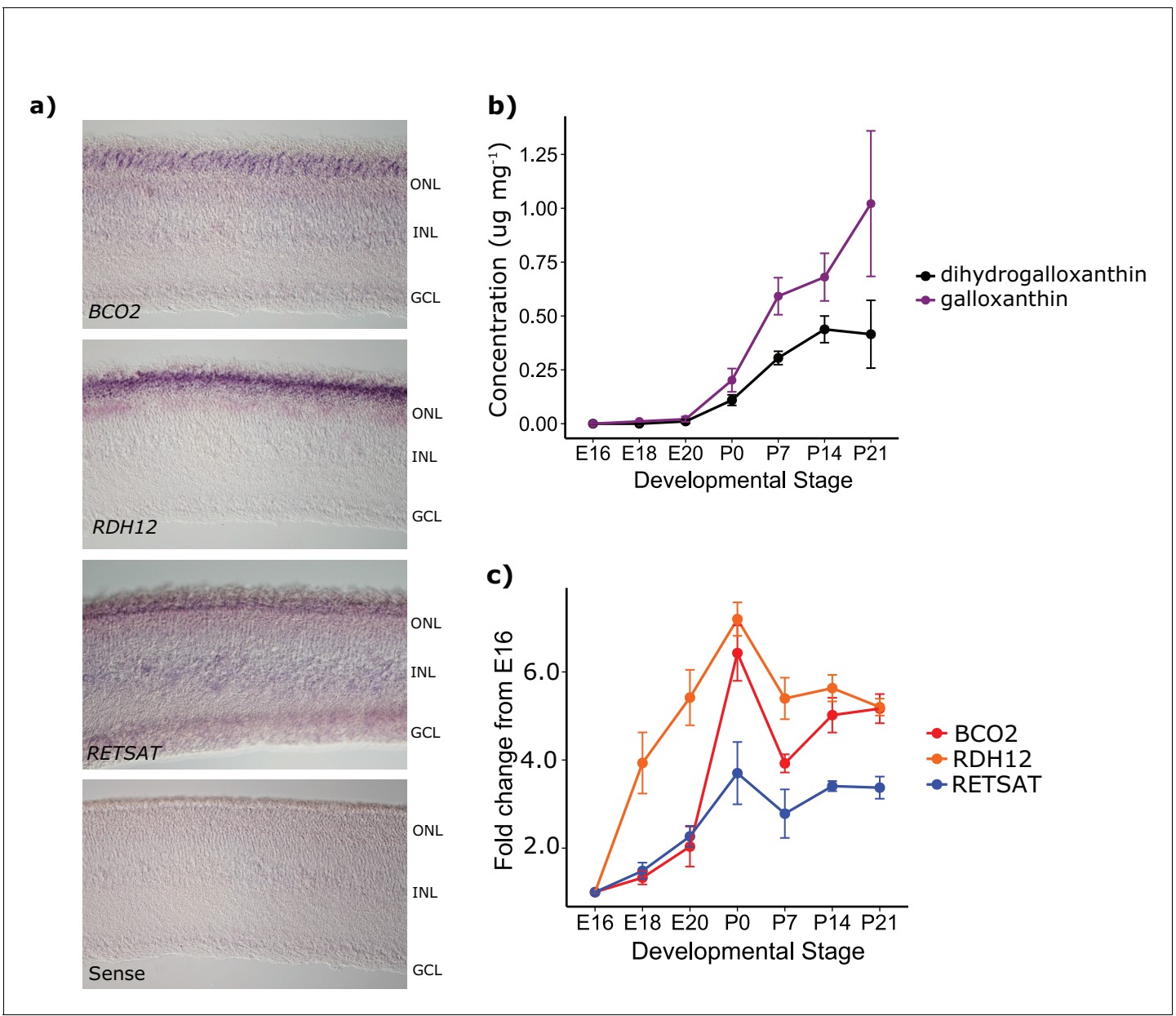

**Figure 5.** The developmental time course and pattern of expression of *BCO2, RDH12*, and *RETSAT* are consistent with their proposed role in apocarotenoid metabolism. (a) In situ hybridization for transcripts of each of the enzymes in the proposed biosynthetic pathway, performed on retinas from newly hatched chickens indicate enzyme expression in the photoreceptor layer of the retina (ONL). No signal was detected in the negative control (*RETSAT* sense probe; bottom panel). The outer nuclear (ONL), inner nuclear (INL) and ganglion cell layers (GCL) are indicated in each section. (b) The concentration ($\mu$g mg$^{-1}$ of protein) of dihydrogalloxanthin and galloxanthin in whole retina extracts from chicken (*Gallus gallus*) increase significantly over the course of development from embryonic day 16 (E16) to 21 days post-hatch (P21). (c) The transcript levels of *BCO2, RETSAT*, and *RDH12* also increase significantly over development. Values are reported as fold-change relative to the earliest time point (E16). Three biological replicates were analyzed at each time point.

The following source data is available for figure 5:

**Source data 1.** Apocarotenoid concentration and transcript expression levels for each biological and technical replicate.

apocarotenoids in the retina at E18, with levels increasing significantly through the course of development (galloxanthin: $F_{6,12}$ = 19.59, p=1.50 x $10^{-5}$, dihydrogalloxanthin: $F_{6,12}$ = 21.00, p=1.03 x $10^{-5}$, **Figure 5b**). Consistent with their hypothesized role in apocarotenoid metabolism, the transcript levels of *BCO2*, *RDH12*, and *RETSAT* increased significantly through the course of development, mirroring the developmental trends in apocarotenoid accumulation (*BCO2*: $F_{6,12}$ = 35.36, p=1.12 x $10^{-7}$, *RDH12*: $F_{6,12}$ = 19.98, p=4.08 x $10^{-6}$, *RETSAT*: $F_{6,12}$ = 15.64, p=1.76 x $10^{-5}$, **Figure 5c**). Taken together, these results indicate that these enzymes are present in the avian retina and are sufficient to produce galloxanthin and dihydrogalloxanthin from dietary carotenoid precursors in cell culture.

## Complementary SWS1 opsin tuning and C-type oil droplet spectral filtering facilitate color discrimination

We have established that the C-type oil droplets are pigmented with galloxanthin and dihydrogalloxanthin and VS and UVS birds modulate the ratio of these pigments to fine-tune the spectral filtering of the SWS2 cone. Next, we sought to investigate if and how these shifts in the spectral filtering of the SWS2 cone influence visual function. We hypothesized that the sensitivity of the SWS2 cone is tuned to facilitate color discrimination and maintain spectral sensitivity across the light spectrum. To test the specific influence of C-type oil droplet filtering on color discrimination, we used a receptor noise-limited (RNL) model of color discrimination to predict the maximum number of discriminable colors for a range of hypothetical visual systems with varying C-type oil droplet spectral filtering (**Vorobyev, 2003**). For this analysis, we used existing measurements of visual physiology from a diverse panel of 7 VS and 11 UVS bird species (**Supplementary file 2**) to predict how idealized spectra of light would stimulate the single cone photoreceptors of each species. With these predicted cone stimulations, we defined a tetrachromatic color space containing all possible spectra that could be encoded by the visual system. We then determined the total number of discriminable colors that could be contained within this space by partitioning it into volumes defined by the minimum distance in color space required to distinguish two spectra. For each species, we varied the C-type oil droplet spectral filtering cutoff ($\lambda_{cut}$) in 6 nm intervals across the spectrum from 370 to 500 nm, while holding all other parameters constant (**Figure 6a,b**), and predicted the total number of discriminable colors at each level of filtering. We identified the optimal C-type spectral filtering as the $\lambda_{cut}$ value that maximized the total number of discriminable colors.

It is important to note that in this analysis we evaluate all possible discriminable colors; however an organism is likely to only encounter a subset of these colors in nature (**Linhares et al., 2008**). In the ideal case, we would limit our analysis to these naturally occurring, behaviorally relevant colors, but the complete set of colors to which a bird is likely to be exposed in nature has yet to be defined for any given species. We therefore chose to analyze visual system performance across the entirety of color space in order to limit the assumptions we made about a given species' 'chromatic ecosystem'.

The RNL model defines discrimination thresholds as a function of the noise deriving from irregularities in the receptor transduction mechanism and the stochastic nature of photon absorbance (i.e., photon-shot noise). The influence of these sources of noise on discrimination depends upon light levels. In bright light, approximately equivalent to sunset or brighter (>10 cd m$^{-2}$), the color discrimination thresholds of birds largely follow Weber's law, indicating that receptor noise scales as a fraction of the stimulus magnitude with small deviations attributable to photon-shot noise (**Olsson et al., 2015**; **Lind et al., 2014a**). In dim light (0.1–10 cd m$^{-2}$), Weber's law no longer holds, and the color discrimination thresholds are limited primarily by photon shot noise (**Olsson et al., 2015**). In very dim light (0.025 to 0.1 cd m$^{-2}$), the avian visual system nears the absolute limit of color discrimination which is likely set by the intrinsic dark noise within the receptors (**Olsson et al., 2015**). To investigate the role of oil droplet spectral filtering across this wide range of potential light conditions, we formulated the RNL model in three ways, each with different noise components: (1) bright light (Weber noise dominant with a component of photon-shot noise), (2) dim light (photon-shot noise dominant), or (3) very dim light (dark noise dominant).

In bright light, the optimal positions of C-type $\lambda_{cut}$ differed significantly between VS and UVS species (independent sample *t*-test, *t* = −7.78, p=2.13 x $10^{-5}$). The optimal positions of the UVS species were short-wavelength shifted (mean ± SD, $\lambda_{cut\ optimal}$ = 425.1 ± 6.5 nm, n = 11) relative to the

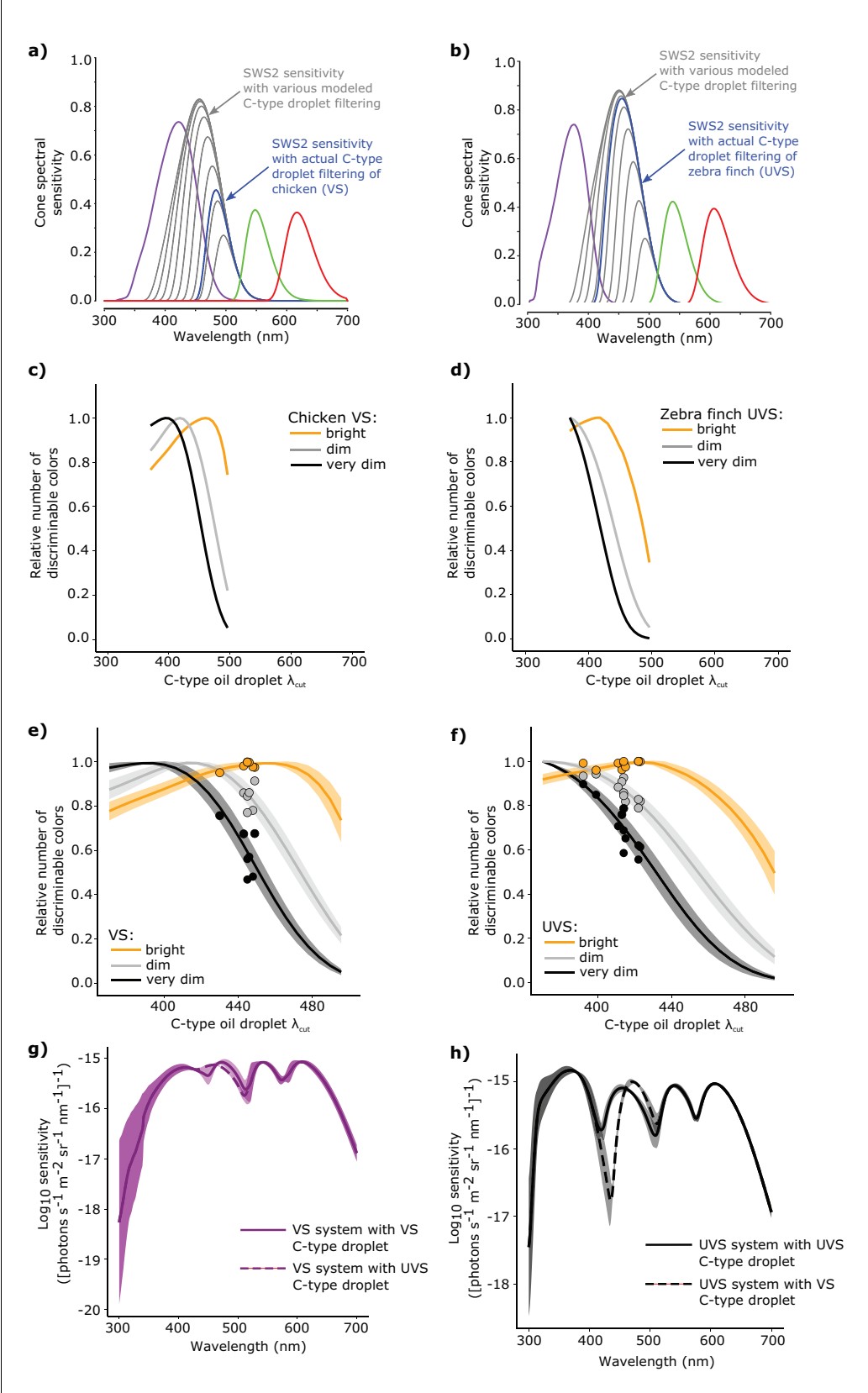

**Figure 6.** The spectral filtering of the C-type oil droplet is nearly optimal for color discrimination in bright light conditions. Examples of the spectral sensitivity values of the LWS (red), MWS (green), SWS2 (blue) and SWS1 cones (purple) of (**a**) chicken (VS) and (**b**) zebra finch (UVS) used to model color

*Figure 6 continued on next page*

*Figure 6 continued*

discrimination. To find the optimal C-type oil droplet spectral filtering, we held all other values constant while varying droplet filtering over a wide range, resulting in shifts in the magnitude and wavelength of peak sensitivity of the SWS2 cone (gray lines). We then predicted the total number of colors that each of the hypothetical visual systems (c–d) could discriminate in bright (yellow lines), dim, (gray lines), and very dim light conditions (black lines). We repeated these analyses for a total of (e) 7 VS species and (f) 11 UVS species, and calculated total number of discriminable colors predicted for each C-type oil droplet $\lambda_{cut}$ value under each of the three lighting conditions. The curves represent the mean ± S.D. total discriminable colors as a proportion of the model maximum for each lighting condition: bright (yellow lines), dim, (gray lines), and very dim conditions (black lines). The points above the curves are the observed $\lambda_{cut}$ of the C-type droplets in each VS and UVS species and the predicted number of discriminable colors relative to the modeled optimum. (g) The mean ± S.D. increment spectral sensitivity of the visual systems of the VS species with the typical oil droplet configuration (solid line) or with mismatched C-type oil droplet filtering typical of UVS species (broken line). (h) The mean ± S.D. increment spectral sensitivity of the visual systems of the UVS species with the typical oil droplet configuration (solid line) or with mismatched C-type oil droplet filtering typical of VS species (broken line).

The following source data and figure supplement are available for figure 6:

**Source data 1.** The number of discriminable colors predicted using the receptor noise-limited model with species-specific ocular media transmittance, spectral sensitivity measures, and varying positions of the C-type oil droplet filtering cutoff.

**Figure supplement 1.** Color discrimination differs significantly between VS and UVS species and lighting conditions.

optima for the VS species ($\lambda_{cut\ optimal}$ = 457.4 ± 9.7 nm, n = 7), a pattern consistent with the actual C-type oil droplet spectral filtering observed within the respective visual systems (*Figure 6c–f*). In the bright light condition, the modeled optima were highly predictive of actual visual system parameters, differing on average by only 2% in the number of discriminable colors between the optimal and actual spectral filtering (*Figure 6e–f*). Overall, the number of discriminable colors predicted for the actual visual system parameters of UVS species ($8.24 \pm 1.77 \times 10^6$, n = 11) were significantly greater than the VS species ($5.18 \pm 1.03 \times 10^6$, n = 7, independent sample *t*-test, $t$ = 4.74, p=0.00021, *Figure 6—figure supplement 1*). This predicted color discrimination advantage of UVS species is consistent with a number of previous color vision modeling analyses (*Vorobyev et al., 1998*; *Cuthill et al., 2000*).

In dim light, our model predicted a significant reduction in the number of discriminable colors compared to the bright-light condition (Bright: $6.95 \pm 2.14 \times 10^6$, Dim: $1.29 \pm 0.45 \times 10^5$, n = 18, paired sample *t*-test, $t$ = 13.94, p=4.34 × $10^{-11}$, *Figure 6—figure supplement 1*). The specific level of dark noise in avian photoreceptors is not known; therefore we did not calculate the absolute number of discriminable colors for the very dim light condition. However, the relative comparison of the models with different levels of C-type oil droplet spectral filtering does not depend on the specific value of dark noise chosen, and we can infer the optimal C-type $\lambda_{cut}$ for the very dim light conditions. In dim and very dim conditions, the optimal C-type $\lambda_{cut}$ differed significantly between VS (dim: 414.6 ± 6.8 nm, very dim: 391.4 ± 9.1, n = 7) and UVS species (dim: 371.6 ± 2.8 nm, very dim: 370.0 ± 0.0, n = 11, independent *t*-test, dim: $t$ = -15.86, p=6.23 × $10^{-7}$, very dim: $t$ = −6.25, p=0.0008). However the predicted optima of spectral filtering in the dim and very dim light conditions differed substantially from the observed values (*Figure 6e–f*). The actual C-type $\lambda_{cut}$ values underperformed the dim and very dim model optima by 14–25%, which was significantly worse than the fit to the bright-light model (mean ± SD proportion of optimal, bright: 0.986 ± 0.016, dim: 0.863 ± 0.057, very dim: 0.745 ± 0.113, n = 18, repeated measures ANOVA, $F_{2,34}$ = 57.44, p=1.25 × $10^{-11}$, *Figure 6f*). These results suggest that the apocarotenoid-based spectral filtering of the SWS2 cone oil droplet is fine-tuned to facilitate color discrimination in bright-light conditions.

Spectral filtering by the cone oil droplets not only fine-tunes the bandwidth and peak sensitivity of the receptor, but also influences its absolute sensitivity to light (*Figure 6a–b*). With the modeling of object color solids (above), we predicted how changes in C-type droplet spectral filtering generally impact color discrimination. For the more specific task of determining how the shifts in droplet filtering affect the discrimination in specific portions of the light spectrum, we modeled the increment threshold spectral sensitivity for the 18 species (*Supplementary file 2*). We tested threshold discrimination for the natural visual system configurations of each species and configurations with mismatched C-type oil droplet filtering typical of a species with the opposite type of visual system

(VS or UVS) (*Lind et al., 2014a*). With this analysis, we determined the minimum light intensity necessary to discriminate a monochromatic light of a given wavelength from a uniform adapting background (the unit of sensitivity is defined as the reciprocal of detection thresholds). When we mismatched the visual systems of VS species with the spectral filtering of UVS-like C-type droplets, our analysis predicted little change in discrimination across the spectrum (*Figure 6g*). In contrast, a mismatch of UVS species visual systems with a VS-like C-type droplet produced a spectral 'blind spot' between 400 and 450 nm, where the discrimination is substantially reduced (*Figure 6h*). These results suggest that the complementary spectral tuning of the SWS2 cone in UVS species, mediated by apocarotenoid-based spectral filtering, is particularly important for the discrimination of spectra that vary in the short-wavelength portions of the light spectrum.

## Discussion

The majority of bird species rely on vision as their primary sensory modality, and color discrimination plays an important role in essential behaviors such as foraging and mate choice (*Bennett and Théry, 2007*). Accordingly, birds have evolved one of the most richly endowed color vision systems among vertebrates (*Goldsmith, 1990*). Here, we have shown how this system relies on the coordinated spectral tuning of distinct cone photoreceptor subtypes (SWS1 and SWS2) via two very different mechanisms (opsin spectral tuning and carotenoid-based filtering). Modeling suggests that these adaptations have served to enhance spectral sensitivity and color discrimination as birds have expanded their visual systems into the UV range.

### Apocarotenoids are important filters of short-wavelength light in the avian retina

We have determined that apocarotenoids are the major spectral filtering pigments within the C-type oil droplet of the SWS2 cone. Apocarotenoids are the products of the oxidative cleavage of 40-carbon carotenoids, which results in truncated molecules with light absorbance spectra that are short wavelength-shifted relative to the parent carotenoid (*Eroglu and Harrison, 2013*; *Moise et al., 2005*). The best understood and arguably most important apocarotenoid in animal biology is retinal (the aldehyde form of vitamin A), which is the product of the symmetrical cleavage of β-carotene by beta-carotene oxygenase 1 (BCO1) (*von Lintig, 2010*). Retinal is the chromophore of the photoreceptor visual pigment, and further oxidation of retinal yields retinoic acid, an essential regulator of gene expression during development (*von Lintig, 2010*). In animals, the functional roles of asymmetric apocarotenoids such as galloxanthin are less well understood, but there is growing evidence that they may be involved in retinoic acid signaling and cancer in mammals (*Eroglu et al., 2012*; *Sharoni et al., 2012*; *Harrison et al., 2012*). Our results demonstrate a specific functional role for asymmetric apocarotenoid products as UV spectral filters in the avian visual system. Oil droplets analogous to the avian C-type droplet are also found in the retinas of turtles and lizards. In addition, the day gecko (*Quedenfeldtia trachyblepharus*) accumulates a UV-absorbing apocarotenoid in its lens (*Röll, 2000*; *Loew et al., 2002*; *Ohtsuka, 1985*). Therefore, UV filtering and photoprotection may be functions of apocarotenoids that are widespread among animals.

In the avian retina, selective changes in the degree of conjugation of carotenoid pigments are an essential mechanism of spectral tuning across the light spectrum. The LWS, MWS, and SWS2 cone photoreceptors each contain a differently colored oil droplet pigmented primarily by one or more carotenoids with a specific degree of conjugation. The LWS cone contains a red droplet pigmented primarily with astaxanthin (13 conjugated double bonds). The MWS cone, has a yellow zeaxanthin-pigmented oil droplet (11 conjugated double bonds), and, as we show here, the SWS2 cone has an oil droplet containing a mixture of dihydrogalloxanthin and galloxanthin (7 or 8 conjugated double bonds, respectively) (*Toomey et al., 2015*). All of these pigments are derived from dietary zeaxanthin through direct accumulation or metabolic modification (*Bhosale et al., 2007*). Thus, the exquisite fine-tuning of avian sensitivity across the light spectrum is achieved through the addition or subtraction of conjugated double bonds in diet-derived carotenoid pigments.

Modulation of the number of conjugated double bonds in the filtering pigments of SWS2 cones represents one instance of a general mechanism of spectral tuning utilized by a diversity of organisms. Many cold-blooded aquatic organisms dynamically shift the sensitivity of their visual system by changing the degree of conjugation of the visual pigment chromophore, retinal (*Bridges, 1972*;

*Enright et al., 2015b*). In contrast to the action of RETSAT (which eliminates a double bond from galloxanthin, thereby causing a short-wavelength shift in C-type oil droplet filtering), in many cold-blooded species the enzyme Cyp27c1 adds a double bond to retinal, thereby increasing the length of the conjugated system and producing a long-wavelength shift in light absorbance (*Enright et al., 2015b*). This shift extends the animal's functional vision well into the near-infrared (*Enright et al., 2015b*). Thus, the addition or subtraction of conjugated double bonds can play a role in tuning the visual system at both ends of the visible spectrum.

## A new role for carotenoid and retinoid metabolizing enzymes

Here, we present evidence that galloxanthin and dihydrogalloxanthin are products of an enzymatic pathway that likely includes BCO2, RDH12, and RETSAT. Each of these enzymes has previously been identified as a carotenoid- or retinoid-metabolizing enzyme (*Kiefer et al., 2001*; *Moise et al., 2004*; *Haeseleer et al., 2002*), but our results suggest broader functional roles for each. BCO2 has primarily been considered a mediator of carotenoid degradation and detoxification (*Lobo et al., 2012a*; *2012b*; *Amengual et al., 2011*). Deficiency of BCO2 in mouse or zebra fish results in an over-accumulation of carotenoids, which induces oxidative stress and cell death (*Lobo et al., 2012a*; *2012b*; *Amengual et al., 2011*). Thus, BCO2 plays an important role in maintaining carotenoid homeostasis through degradation, and its products do not typically accumulate in tissue. In the avian retina, in contrast, these apocarotenoid products of BCO2 are retained, accumulate to high concentrations, and play an important role in spectral tuning. Thus, birds appear to have repurposed a carotenoid-degrading enzyme to fine-tune color vision.

RDH12 is the most highly expressed member of the short-chain dehydrogenase/reductase (SDR) family in avian photoreceptors (*Enright et al., 2015a*). In mammalian systems this enzyme has been shown to efficiently catalyze the reduction of retinal to retinol, playing a key role in the visual cycle (*Belyaeva et al., 2005*). Our results indicate that avian RDH12 acts on substrates other than retinoids and readily reduces 3-Hydroxy-10'-apo-β-caroten-10'-al to galloxanthin. We recognize, however, that there are additional SDR family members expressed in avian photoreceptors (*Enright et al., 2015a*) that may also have the ability to act upon apocarotenal substrates.

We have proposed that the 11',12' saturation of galloxanthin is mediated by RETSAT, an enzyme known to carry out an analogous saturation of retinol to produce 13,14-dihydroretinol (*Moise et al., 2004*). In this study, we found that avian RETSAT accepts a broader range of substrates than previously reported for orthologs from other species, and that it readily saturates the apocarotenoid galloxanthin to produce dihydrogalloxanthin. What remains to be determined is how the expression and activity of RETSAT is modulated between VS and UVS species to produce the specific galloxanthin:dihydrogalloxanthin ratios found in individual species' visual systems.

## Shifts in spectral filtering are required to unlock the full adaptive potential of the VS to UVS switch in avian color vision

Changes in the filtering of the C-type oil droplet between VS and UVS bird species shift the peak sensitivity of the SWS2 cone photoreceptor to maintain even spectral separation from the adjacent SWS1 and MWS photoreceptors. Interestingly, perfect spectral separation among all of the single cone photoreceptor subtypes would also require a blue-shift in the sensitivity of the MWS cone in UVS species. However, for the species measured to date, there are no systematic differences between VS and UVS species in the positioning of the spectral sensitivity of the MWS cone. This suggests that the spectral tuning of MWS may be mechanistically constrained or its precise tuning has a relatively limited impact on visual function.

Our models of color discrimination indicate that shifts in C-type oil droplet filtering bring the VS and UVS visual systems to a near-optimal configuration for color discrimination in bright light. The short-wavelength shift of C-type droplet filtering in UVS species also maintains sensitivity across the visual spectrum and eliminates potential spectral 'blind spots'. These co-adaptations appear to have been driven specifically by selection for color vision under bright-light conditions. In dim and very dim light, our models predicted a substantial mismatch between the optimal C-type droplet $\lambda_{cut}$ values and actual measured values, and an overall decline in color discrimination performance. This is not necessarily surprising, because the species we examined are all primarily active in bright daylight conditions and unlikely to be adapted for low-light vision. Interestingly, nocturnal species (e.g. owls)

have been reported to have pale oil droplets that lack carotenoids (*Bowmaker and Martin, 1978*). This decrease in oil droplet pigmentation may represent an adaptation for increased cone sensitivity in dim light which is consistent with our predictions for the optimization of discrimination in low-light conditions (*Bowmaker and Martin, 1978*). Taken together, our results support the hypothesis that the complementary SWS1 opsin spectral tuning and C-type oil droplet filtering are adaptations for enhanced color discrimination in bright-light conditions and sustained sensitivity across the visual spectrum.

The shift from a VS to UVS visual system is also correlated with an increase in the UV transmittance of the ocular media (*Lind et al., 2014b*). The filtering of shortwave-length light by the ocular media determines the short wavelength limit of the SWS1 cone sensitivity, in much the same way as the C-type droplet filters the spectrum of the SWS2 cone. The increase in UV transmittance in UVS species has marginal effects on color discrimination, but substantially increases the absolute sensitivity of the SWS1 cone, which improves color discrimination in low-light conditions (*Lind et al., 2014b*). The mechanisms determining ocular media transmittance are not known at this time but may involve the accumulation of pigments and/or changes in lens and cornea proteins. Thus, the transition from VS to UVS vision involves not only the well characterized changes in the SWS1 opsin, but also complementary shifts in the filtering of both the ocular media and the C-type droplet of the SWS2 cone. Additionally, it has recently been shown that the optics of avian cone photoreceptor subtypes are adapted to complement the specific pigmentation and refractive properties of the cone oil droplets and facilitate the transmission of light (*Wilby et al., 2015*). Thus, it appears that a complex suite of optical adaptations are required to obtain maximal benefit from the VS-to-UVS shift.

Our results highlight a degree of complexity to avian spectral tuning that is not always discussed in studies of avian visual ecology. The spectral tuning of the SWS1 opsin can be readily inferred by gene sequencing, and is frequently used to categorize species as VS or UVS (*Ödeen and Håstad, 2003*; *2013*; *Hauser et al., 2014*). However, the presence of a UV-shifted SWS1 opsin does not guarantee that the complementary changes in ocular media and oil droplet filtering are present. Despite this fact, generalized sets of visual system parameters for VS or UVS visual systems are often used to model how a specific bird species sees color signals, under the untested assumption that the visual systems are relatively invariant among VS or UVS species (e.g. (*Avilés and Soler, 2009*; *Friedman and Remeš, 2015*)). Our results indicate that an exclusive focus on opsin-based spectral tuning may offer an incomplete view of the spectral adaptations of the avian visual system. For example, a recent study of warblers from distinct genera and visual ecologies observed little variation in SWS1 and SWS2 opsin spectral sensitivities (*Bloch et al., 2015*). However, adaption in these visual systems could be mediated through changes in filtering of the ocular media and oil droplets that are not apparent from measures of the opsins alone.

## Conclusion

Color vision is essential to birds, and here we have shown how distinct mechanisms of spectral tuning in different cell types are coordinated to optimize color discrimination. The VS-to-UVS switch in avian vision has become a model system for the integrated study of molecular, sensory, and signaling evolution with a specific focus on the SWS1 opsin (*Ödeen and Håstad, 2003*; *Shi et al., 2001*; *Friedman and Remeš, 2015*; *Håstad et al., 2005*; *van Hazel et al., 2013*; *Ödeen et al., 2012*). Our results present a more comprehensive picture of the functional adaptations that underlie this switch and reveal that the full potential of UVS vision is only realized when there are complementary changes in spectral filtering of the C-type oil droplet of the SWS2 cone, mediated by the selective accumulation of apocarotenoid pigments.

## Materials and methods

### LC-MS analysis of retinal apocarotenoids

Whole retinas from chickens (21 days post-hatch) were extracted three times with 500 µl of hexane: tert-butyl methyl ether 1:1 (vol:vol). The extracts were combined, dried under a stream of nitrogen, and saponified with 0.2M sodium hydroxide in methanol at room temperature, in the dark for six hours, as detailed in (*Toomey and McGraw, 2007*). We re-extracted the saponified carotenoids with

2 ml of hexane:tert-butyl methyl ether 1:1 (vol:vol) and dried them under nitrogen. We separated the apocarotenoids on a Waters 2695 gradient HPLC separation module (Waters Corp.,Milford, MA, USA) equipped with an autoinjector, a 996-photodiode array detector (PDA), and a C18 Symmetry column (75 mm × 4.6 mm i.d., 3.5 $\mu$m) (Waters Corp.). We used a mobile phase gradient beginning with 20:80 water:methanol (v:v) up to 100% methanol over the course of 5 min. Detection was performed using a quadrupole/time-of-flight mass spectrometer (QTof Premier, Micromass Limited, Manchester, UK) equipped with an electrospray ionization (ESI) source operated in both positive and negative modes of polarity. ESI conditions included a capillary voltage of 3.2 kV for positive mode, 2.8 kV for negative, cone voltage of 35 V, ion guide at 1 V, source temperature of 100°C, and nitrogen desolvation gas temperature of 400°C flowing at 600 L/h. We compared the observed apocarotenoids to authentic standards of galloxanthin and 3-OH-β-apo-12′-carotenol purchased from CaroteNature GmbH (Ostermundigen, Switzerland).

## Prediction of $\lambda_{max}$ for monosaturated forms of galloxanthin

To predict the wavelength of maximum absorbance ($\lambda_{max}$) for all possible monosaturated forms of galloxanthin, we used the Woodward–Fieser rule for molecules with $\leq$4 conjugated double bonds and the Fieser-Kuhn rule for molecules containing >4 double bonds (*Kalsi, 2007*). According to the Woodward–Fieser rule, $\lambda_{max}$ values were calculated as follows for molecules with the base diene containing an endocyclic bond:

$$\lambda_{max} = 214 + 30n + 5M + 5R_{exo}$$

or for molecules without an endocyclic bond:

$$\lambda_{max} = 215 + 30n + 5M$$

In these calculations, $n$ is the number of conjugated double bonds in addition to the base diene, $M$ is the number of alkyl substituents, and $R_{exo}$ is the number of exocyclic double bonds in the base diene. According to the Fieser-Kuhn rule, $\lambda_{max}$ values were calculated as follows:

$$\lambda_{max} = 114 + 5M + n(48.0 - 1.7n) - 16.5\,R_{endo} - 10R_{exo}$$

where $n$ is the total number of conjugated double bonds, $M$ is the number of alkyl substituents, $R_{endo}$ is the number of endocyclic double bonds in the molecule, and $R_{exo}$ is the number of exocyclic double bonds in the molecule.

## Derivatization and ozonolysis experiments to confirm 11′,12′ saturation of dihydrogalloxanthin

Approximately 200 mg (1.96 mmol) of 5-hydroxy-2-pentanone (Sigma-Aldrich, St. Louis, MO, USA) was dissolved in $CH_3CN$ and 750 mg (4.04 mmol) of 4-nitrobenzoyl chloride (compound **1** in *Figure 2—figure supplement 4*; Sigma-Aldrich), and 350 µL of pyridine were added, and the mixture was stirred under argon at room temperature overnight. The resulting suspension was diluted with ethyl acetate and washed with 1N HCl, water, saturated $NaHCO_3$ solution, saturated NaCl solution, dried (anhydrous $MgSO_4$), filtered and concentrated to give a residue that solidified under vacuum. A portion of this solid was purified by silica gel preparative thin-layer chromatography (5% $CH_3OH$/$CH_2Cl_2$) to give a white solid with the expected properties for model compound **2** (*Figure 2—figure supplement 4*): [1]H NMR (400 MHz, $CD_3COCD_3$) δ 1.98 (m, 2H), 2.07 (s, 3H), 2.65 (t, 2H), 4.32 (t, 2H), 8.22 (d, 2H), 8.32 (d, 2H); UV ($CH_3OH$) $\lambda_{max}$ 257 nm; HPLC $t_R$ = 6.8 min (0.8 mL/min of 70% $CH_3OH$/$H_2O$ on a 250 x 4.6 mm Polaris C18 column). Negative ion APCI MS showed the expected molecular ion for $C_{13}H_{12}NO_5$ at *m/z* 251.08102. The ozonolysis products were chromatographed using an Eclipse plus HD C18 (2.1 x 50 mm, 1.8 µm; Agilent Technologies, Santa Clara, CA) with a 0.1%(v/v) formic acid in water versus 0.1%(v/v) formic acid in methanol mobile phase gradient at 0.3 mL/min and 40°C column temperature. From 0–3 min a gradient was applied from 80/20 A/B to 0/100, held for 1 min, and then re-equilibrated over 1 min to initial conditions. Eluent was interfaced via an APCI probe in negative mode with a 6550 QTOF (Agilent) with 40 µA corona, 3 kV vcap, 40 psi nebulizer, 15 L/min drying gas at 290°C, and 500°C vaporizer. MS and MS/MS data were acquired at 2 Hz from 100–1700 m/z, and MS/MS spectra were collected at 10 eV.

The HPLC isolated fraction from the retina of the house finch (*Haemorhous mexicanus*), a UVS species, containing approximately 2 µg of the unknown apocarotenoid (**3** in *Figure 2—figure supplement 4*), was evaporated to dryness under a stream of dry argon, dissolved in dry CH$_3$CN, and compound **1** (7.5 mg, 0.04 mmol, *Figure 2—figure supplement 3*), and 35 µL of pyridine added. The mixture was stirred at room temperature and worked up as for the model compound above to give an off-white solid. This residue was dissolved in CH$_2$Cl$_2$, 10 mg of *N*-methylmorpholine-*N*-oxide (0.085 mmol; Sigma-Aldrich) added and the solution was saturated with ozone (O$_3$) generated from dry O$_2$ using a Welsbach (Philadelphia, PA, USA) laboratory ozonator, model T-816 (the apocarotenoid solution containing presumed compound **4** (*Figure 2—figure supplement 4*) first decolorized and then turned blue when saturated with O$_3$). An equal volume of 25 mM phosphate buffer, pH 6, was added, the two layers shaken together (*Schwartz et al., 2006*), and the CH$_2$Cl$_2$ layer removed, dried (anhydrous MgSO$_4$) filtered and concentrated. Analysis of the residue by HPLC as for the model compound **2** showed a band at the comparable retention time (see above). Further analysis of this sample by LC-MS showed the presence of a component identical to model compound **2** and, importantly, no evidence for the smaller fragment (C$_9$H$_7$NO$_5$), which would have formed by apocarotenoid ozonolysis had there been an 11',12' double bond present to oxidatively cleave.

## HPLC analyses of retinal apocarotenoid composition

Avian eye samples (*Supplementary file 1*) were collected from various locations in North and South America by the Museum of Southwestern Biology (University of New Mexico, Albuquerque, NM) or as part of previous studies (*Toomey et al., 2015*; *Smith et al., 2011*; *Toomey and McGraw, 2009*). The eyes or isolated retinas were frozen immediately following collection and stored at −80°C. For analysis, we thawed the eyes on ice, dissected the retinas into 500 µl of phosphate buffered saline (PBS), and homogenized them with a plastic pestle in a 1.5 ml centrifuge tube. To determine protein concentration of the homogenate we used a bicinchoninic acid assay (cat. no. 23250, Thermo Scientific, Rockford, IL, USA). We then extracted 500 µl of each homogenized retina with 500 µl of hexane:tert-butyl methyl ether 1:1 (vol:vol), repeated this extraction three times, and combined and dried the extracts under a stream of nitrogen. We saponified and dried the extracts as described above, resuspended this material in 200 µl of methanol:acetonitrile 1:1 (vol:vol) and injected 100 µl into an Agilent 1100 series HPLC fitted with a YMC carotenoid 5.0 µm column (4.6 mm × 250 mm, YMC Inc. Allentown, PA). We eluted the carotenoids with a gradient mobile phase consisting of acetonitrile:methanol (50:50) for 9 min, followed by a ramp up to acetonitrile:methanol:dichloromethane (44:44:12) (vol:vol:vol) from 9–11 min, isocratic through 21 min, then a ramp up to acetonitrile:methanol:dichloromethane (35:35:30) from 21–26 min followed by isocratic conditions through 35 min. The column was held at 18°C, and the flow rate was 1.0 ml/min throughout the run. We monitored the samples at 400 nm and identified galloxanthin by comparison to an authentic standard purchased from CaroteNature GmbH (Ostermundigen, Switzerland).

## SWS1 opsin sequence analysis

To determine the spectral tuning of the SWS1 opsin, we referenced published sequence data when available (*Supplementary file 1*) or sequenced the transmembrane region 2 (TMRII) of the opsin from genomic DNA. For sequencing, we extracted genomic DNA from extra ocular muscle tissue with the DNeasy Blood and Tissue kit (QAIGEN) following manufacturer's instructions. To amplify portions of TMRII, we used previously publish degenerate primers: su149a, su193a, su306b, su353b, and su396b in various combinations (*Ödeen and Håstad, 2003*; *2009*). The PCR reactions contained 1–16 ng/µl of genomic DNA, 1 unit of goTaq polymerase and reaction buffer (Promega), 0.5 µM of forward and reverse primers, 0.2 mM dNTPs, and 1.5 mM MgCl$_2$. We ran the reactions on a BioRad C1000 touch thermocycler with conditions: 5 × (30 s at 94°C, 30 s at 54°C, and 1 s at 72°C) followed by 38 × (30 s at 94°C, 30 s at 54°C, and 5 s at 72°C) and 10 min at 72°C. The PCR products were purified on a 2% agarose gel, extracted with QIAquick Gel Purification kit (Qiagen) and the resulting products were sequenced by the Protein Nucleic Acid Chemistry Laboratory (PNACL) at Washington University using the same primers as used in the initial PCR reactions. We translated the resulting DNA sequence and identified the tuning sites at AA86, AA90, and AA93 and inferred VS vs UVS tuning following published analyses (*Hauser et al., 2014*; *Wilkie et al., 2000*).

We then examined the relationship between SWS1 opsin tuning (VS/UVS) and the relative abundance of the different apocarotenoids in the retinas of these species. To analyze the data in a phylogenetic context, we generated 100 phylogenies for all sampled bird species using the resources at http://birdtree.org/subsets/ (*Jetz et al., 2012*), with the deep phylogenetic relationships, or backbone, guided by Hackett et al. (*Hackett et al., 2008*) and adjusted to conform with Jarvis et al. (*Jarvis et al., 2014*). For each of our two traits of interest, we quantified phylogenetic inertia using Blomberg's K and Pagel's λ, we tested Brownian motion versus Ornstein Uhlenbeck (OU) models of evolution (Brownian motion was favored by AICc in all instances), and we estimated ancestral states using maximum likelihood. We tested the relationship between VS/UVS and the ratio of dihydrogalloxanthin to total apocarotenoids with a phylogentic generalized linear model. The results of these analyses were nearly invariable among the 100 tree topologies, indicating that there was no sensitivity to phylogenetic uncertainty. Analyses utilized *R* packages *ape, nlme,* and *picante* (*Kembel et al., 2010*; *Paradis et al., 2004*; *Pinheiro et al., 2014*).

## Microspectrophotometry and spectral curve fitting to determine C-type droplet composition

The apocarotenoid compositions of the C-type droplets in chicken (VS), mallard (VS), zebra finch (UVS), and house finch (UVS) retinas were determined by collecting detailed absorbance spectra from expanded individual C-type droplets with microspectrophotometry and fitting these with the pure spectra of the seven major carotenoids and apocarotenoids in the retina. These methods are detailed in *Toomey et al., (2015)*.

## HEK293 cell assay of apocarotenoid metabolizing enzyme function

To test the activity of the enzymes in our proposed apocarotenoid metabolic pathway, we cloned full-length transcripts of *BCO2* and *RDH12* from 21 day-old chicken retina cDNA and *RETSAT* from adult zebra finch retina cDNA using the primers listed in *Supplementary file 3a*. We subcloned these products into the first position of a bicistronic Tet-inducible expression vector pTre3G-IRES-dsRed that we modified to include *Nde*I and *Bsr*GI restriction sites and a C-terminal c-Myc tagging sequence. We confirmed the sequence of each gene by Sanger sequencing (PNACL). We cultured HEK293 cells (ATCC, CRL-1573) following the distributor's recommended protocols for up to 25 passages. For the enzyme assays, we cultured the cells in 6-well plates (each well 9.6 cm$^2$) to 80% confluency and transfected the enzyme expression constructs along with a vector containing a CMV promoter driving the expression of the reverse tetracyclin-controlled transactivator using the TransIT reagent (Mirus Bio, Madison, WI). Twenty four hours after transfection, we induced protein expression by adding doxycycline to the media at a concentration of 1 µg/ml. We purified zeaxanthin from a bacterial culture transformed with the zeaxanthin-producing plasmid pAC Zeax, a gift of Francis Cunningham (University of Maryland, College Park, MD) (*Cunningham and Gantt, 2007*). To produce supplemented media, we combined 5.5 µg of purified zeaxanthin, with 1 µl of Tween 40 (Sigma, P1504), and 1 ml of complete media. At 24 hr post-induction, we removed all media from the wells and replaced it with the zeaxanthin-supplemented media. We incubated the cells overnight (16 hr), then collected cells and media, added 1 ml of ethanol, and then extracted with 2 ml of hexane: tert-butyl methyl ether. We collected the solvent fraction, dried it under a stream of nitrogen, resuspended the extracted carotenoids in 200 µl of methanol:acetonitrile 1:1 (vol:vol), and injected 50 µl into the Agilent 1100 HPLC system described above.

We eluted the carotenoids with a gradient mobile phase consisting of acetonitrile:methanol:dichloromethane (44:44:12) (vol:vol:vol) from 1–11 min, then a ramp up to acetonitrile:methanol:dichloromethane (35:35:30) from 25 min, followed by isocratic conditions through 30 min. The column was held at 30°C and the flow rate was 1.2 ml/min throughout the run. We monitored the samples at 400 nm and identified galloxanthin by comparison to an authentic standard purchased from CaroteNature GmbH (Ostermundigen, Switzerland). We identified dihydrogalloxanthin by comparing absorbance spectra and retention times to measurements of this compound in retina samples. We putatively identified 3-OH-β-apo-10′-carotenal by comparison to published reports of this apocarotenoid (*Mein et al., 2010*).

## In situ hybridization for apocarotenoid metabolizing enzymes transcripts in the avian retina

We cloned probe templates for *BCO2, RDH12*, and *RETSAT* from the retinal cDNA of 21 day-old chicken with PCR primers that included *Eco*RI or *Xho*1 restriction sites (*Supplementary file 3b*). We subcloned the PCR products into the BlueScript vector pBSK+ at the *Eco*RI or *Xho*1 restriction site and confirmed the sequences by Sanger sequencing (PNACL). We then generated probe templates by PCR with T3 or T7 primers and generated digoxigen-labeled probes from these templates following established methods (*Enright et al., 2015a*; *Trimarchi et al., 2007*). We harvested whole eyes from P0 chickens and fixed them overnight (~16 hr) in 4% paraformaldehyde in 1x PBS. We then embedded the in Tissue-Tek OCT compound (Sakura, Torrance, CA) and cut 12 μm horizontal sections through the center of the eye. We carried out in situ hybridization on these sections as previously described (*Enright et al., 2015a*) and imaged the hybridized sections at 400x on an Olympus BX-51 microscope.

## Developmental time course of apocarotenoid accumulation and enzyme transcript expression in the developing chicken retina

To measure the accumulation of apocarotenoids and the transcript levels of *BCO2, RDH12*, and *RETSAT* in developing chicken retinas, we purchased specific pathogen-free eggs from Charles River Laboratories (North Franklin, CT), incubated eggs, and reared chicks to specific developmental stages. Chicks were fed a carotenoid-rich diet (Start & Grow, Purina Mills, St. Louis, MO). We harvested retinas from the embryos (E) at days 16, 18, and 20 of incubation and chicks at days 0, 7, 14, and 21 post-hatch (P). The birds were euthanized by carbon dioxide asphyxiation following an approved protocol (Washington University ASC protocol #20140072). We collected three biological replicates of retina tissue at each time point for each analysis. We measured apocarotenoids levels by HPLC as described above and used multiple analysis of variance (MANOVA) to compare the change in apocarotenoid levels over time.

For qPCR analysis, we harvested retinas and stored them in Trizol reagent (Life Technologies, Carlsbad, CA) for up to six months at -80°C. We then extracted RNA from the samples following manufacturer's recommendations (Trizol reagent, Life Technologies, Carlsbad, CA). We generated cDNA from ~1 μg of RNA using Superscript IV reverse transcriptase (Life Technologies, Carlsbad, CA). We selected primers flanking the last exon/exon junction of *BCO2, RDH12, RETSAT, and GAPDH* (*Supplementary file 3c*). To test primer efficiency we assayed a 1:10 dilution series of P21 chicken retina cDNA. The primer efficiencies ranged from 98% to 106% at the threshold ($C_t$) levels used for quantification in the time course analyses. We assayed each time point and biological replicate in triplicate using Sybr Green PCR master mix (Life Technologies, Carlsbad, CA) with an Applied Biosystems StepOne real time PCR system (Carlsbad, CA). We normalized enzyme transcript expression relative to *GAPDH* expression. We then averaged the technical replicates for each biological replicate and compared expression levels ($\Delta C_t$) over time for each gene with analysis of variance (ANOVA).

## Modeling the object-color solid and the number of discriminable colors

To evaluate the role of the spectral tuning of C-droplet, we used mathematical models of color discrimination. Colors can be represented as loci within *n*-dimensional color space, where dimensionality corresponds to the number of spectrally different photoreceptors encoding the colors. The coordinate axes represent the quantum catch, *Q*, of these photoreceptors and are given by:

$$Q_i = \int_{\lambda_{min}}^{\lambda_{max}} R_i(\lambda)S(\lambda)I(\lambda)d\lambda \qquad (1)$$

where *i* is receptor type (*i* = UVS/VS, SWS, MWS, LWS), $\lambda_{max}$ and $\lambda_{min}$ are the upper and lower wavelength limits of the visible spectrum, which are set to 700 nm and 300 nm respectively, $R_i(\lambda)$ is receptor sensitivity, $S(\lambda)$ is the reflectance spectrum and $I(\lambda)$ is the illumination spectrum.

Receptor sensitivity was determined as:

$$R_i(\lambda) = v_i(\lambda)p_i(\lambda)o(\lambda) \qquad (2)$$

where $v_i(\lambda)$ and $p_i(\lambda)$ represent visual pigment sensitivity and oil droplet transmittance respectively, calculated from published data used with a visual pigment template (*Govardovskii et al., 2000*) and an oil droplet model (*Hart and Vorobyev, 2005*), and $o(\lambda)$ is the transmittance of the ocular media (*Supplementary file 2*).

As a measure of performance, we used the number of discriminable colors within the object-color solid. The color solid is the volume within color space occupied by all possible object colors given by reflectance spectra, $S$, that do not exceed one, thus $0 \leq S(\lambda) \leq 1$. The boundaries of the color solid are formed by the color loci of optimal color stimuli that are imaginary reflectance spectra with only two possible values, 0 or 1, and with $n$-1 transitions ($n$ represents color space dimensionality). The solid defined by optimal color stimuli has two apices corresponding to an ideal black ($S(\lambda) \equiv 0$) and an ideal white surface ($S(\lambda) \equiv 1$), while the solid shape depends on the spectral tuning of the photoreceptors. Generally, solid volume is larger for photoreceptors with less spectral overlap (the solid completely fills color space for the ideal case of completely non-overlapping photoreceptors).

Each color locus in color space is surrounded by a 'confusion-volume', $u$, containing the loci of all colors that cannot be discriminated from the central locus. The distance between the central locus and the surface of the confusion-volume represents threshold difference, which is a property that varies with position and direction in color space. The total number of discriminable colors for an object color-solid of volume, $V$, is given by:

$$N = \int \frac{1}{u} \prod_{i=1}^{i=n} dQ_i \tag{3}$$

and integration is carried out over the complete solid volume. We assume that thresholds are determined by photoreceptor noise, which has proven a successful approach in modeling of color discrimination (*Olsson et al., 2015*). Relevant for our modeling is intrinsic photoreceptor noise that comes from irregularities in the transduction mechanism and can be defined, in bright light conditions, by a constant Weber fraction (0.2:0.14:0.14:0.1 for UVS/VS:SWS:MWS:LWS), $\omega$;

$$\delta Q_i = \omega_i Q_i \tag{4}$$

and photon-shot noise that originates in the stochastic nature of photon absorption:

$$\delta Q_i = \sqrt{Q_i} \tag{5}$$

In very dim light, dark noise predominates. This noise does not depend on the light signal:

$$\delta Q_i = d_i \tag{6}$$

We consider three cases corresponding to different levels of illumination. In very dim light, we assume that the dark noise limits discrimination (*Equation 6*). The absolute level of dark noise in avian cones is unknown ([*Olsson et al., 2015*], and references therein). Here, we have assumed it to be 50 events per integration time, but this is irrelevant as we are only interested in relative performance of visual systems with different oil droplet configurations. In dim light, thresholds are dominated by shot noise (*Equation 5*). In bright light, we assume that photon-shot noise and the intrinsic photoreceptor noise are important:

$$\delta Q_i = \sqrt{Q_i + \omega_i^2 Q_i^2} \tag{7}$$

From the photoreceptor noise, we can estimate the confusion-volume by:

$$u = c \prod_{i=1}^{i=n} \delta Q_i \tag{8}$$

The parameter, $c$, depends on the relationship between the threshold and noise but is omitted from the calculations as we assume that the threshold criterion is invariant. *Equation 8* describes a tesseract in the four-dimensional color space of birds that is proportional to the real confusion volume, which generally has a smooth (four-dimensional) ellipsoidal shape. The object-color solid is defined by surface reflectance spectra. The number of discriminable colors within the solid is therefore dependent on how these surfaces are illuminated. We assume that the illuminant is standard

daylight, d65 (*Wyszeski and Stiles, 1982*), with a normalized spectrum, $j(\lambda)$, and a maximum intensity of $T$ (assuming a green illumination spectrum given by vegetation gives very similar results to those reported here). We assumed levels of illumination equivalent to a quantum catch of the LWS cone of 100,000, 100, and 10 photons sec.$^{-1}$ for the bright, dim, and very dim conditions, respectively. We describe the quantum efficiency of photoreceptor $i$ for the illuminating spectrum by:

$$k_i = \int_{\lambda_{min}}^{\lambda_{max}} R_i(\lambda)j(\lambda)d\lambda \qquad (9)$$

and we can reformulate quantum catch (*Equation 1*) as:

$$Q_i = Tk_ix_i \qquad (10)$$

where $x$ is given by:

$$x_i = \int_{\lambda_{min}}^{\lambda_{max}} r_i(\lambda)S(\lambda)j(\lambda)d\lambda \qquad (11)$$

and

$$r_i(\lambda) = \frac{R_i(\lambda)}{k_i} \qquad (12)$$

We normalize the scaling factors describing photoreceptor quantum efficiencies to the LWS cone, i.e. $k_{LWS} = 1$, and we fix the spectral sensitivity of the UVS, MWS and LWS cones so that both $k_{UVS/VS}$ and $k_{MWS}$ are kept invariant. For each species we repeat the calculations of the color solids using different C-droplet positions, which yield different spectral variants of the SWS cone and thereby different values for $k_{SWS}$. This approach is valid because we determine the optimal C-droplet position for the maximal number of discriminable colors, while absolute numbers are irrelevant. The shape of the color solid can now be represented by a 'shape factor', $G$. The shape factor is calculated differently depending on the origin of noise. In the case of very dim light:

$$G = \int \prod_{i=1}^{i=n} dx_i \qquad (13)$$

and the total number of discriminable colors is given by:

$$N = \frac{T^n}{c\prod_{i=1}^{i=n} d_i} G \prod_{i=1}^{i=n} k_i \qquad (14)$$

If the fluctuations of absorbed photons limit the color discriminability, then

$$G = \int \frac{1}{\prod_{i=1}^{i=n} \sqrt{x_i}} \prod_{i=1}^{i=n} dx_i \qquad (15)$$

And the total number of discriminable colors is given by

$$N = \frac{1}{c} T^{\frac{n}{2}} G \prod_{i=1}^{i=n} \sqrt{k_i} \qquad (16)$$

In bright light, when both the Weber noise and the shot noise play role, the shape factor depends on the light intensity;

$$G = \int \frac{1}{\prod_{i=1}^{i=n} \sqrt{x_i + \omega_i^2 Tk_ix_i^2}} \prod_{i=1}^{i=n} dx_i \qquad (17)$$

and the total number discriminable colors are calculated using *Equation 16*.

More details of the theory behind object-color solids and how they can be computed are found in (*Vorobyev, 2003*; *Wyszeski and Stiles, 1982*). We compared the optimal predicted C-type droplet filtering and the predicted number of discriminable colors between VS and UVS species with independent sample $t$-tests. We compared the optimal predicted C-type droplet filtering and the

predicted number of discriminable colors between bright and dim light conditions with paired sample $t$-tests.

## Modeling the increment spectral sensitivity function

We modeled the increment spectral sensitivity function using a commonly adopted receptor noise-limited model (RNL) of color discrimination (*Vorobyev and Osorio, 1998*). The RNL-model accounts for chromatic mechanisms, ignores achromatic signals, and postulates that detection is limited by photoreceptor noise. The model accounts for color opponent mechanisms. However, these can be left unspecified (as long as all receptors are compared) because discrimination is limited by receptor noise and not how photoreceptor output is combined (*Vorobyev and Osorio, 1998*). Here follows a short model description while a comprehensive model outline was described earlier (*Vorobyev et al., 2001*).

The increment spectral sensitivity function describes detection of monochromatic light superimposed on a large adaptive background, where sensitivity is expressed as the inverse of the detection thresholds. For large and static stimuli, detection is mediated by color vision. We account for background adaption by defining photoreceptor quantum catch in relation to the quantum catch of the background. From *Equation 1* it follows that the quantum catch of background is given by $Q_i = \int_{300}^{700} R_i(\lambda) I_b(\lambda) d\lambda$, where $I_b(\lambda)$ is the background spectrum, and for monochromatic stimuli (of wavelength $\lambda$) the difference in quantum catch is given by $\Delta Q_i = R_i(\lambda) I_{t\lambda}$, where $I_t(\lambda)$ is the intensity of test monochromatic light of the wavelength $\lambda$. Therefore the relative difference of quantum catches can be calculated as:

$$\Delta q_i = \frac{\Delta Q_i}{Q_i} = \frac{R_i(\lambda) I_{t\lambda}}{\int_{300}^{700} R_i(\lambda) I_b(\lambda) d\lambda} \tag{18}$$

Note that for small $\Delta q_i$ the relative difference of quantum catches can be defined also as the difference of logarithms of quantum catches, i.e. $\Delta q_i = \frac{\Delta Q_i}{Q_i} = \Delta \ln(Q_i)$ (*Vorobyev et al., 2001*). For background, we use a standard daylight spectrum (d65; (*Wyszeski and Stiles, 1982*). The chromatic contrast between the stimulus and the background, $\Delta$S S, is calculated as:

$$(\Delta S)^2 = \frac{(\omega_1 \omega_2)^2 (\Delta q_4 - \Delta q_3)^2 + (\omega_1 \omega_3)^2 (\Delta q_4 - \Delta q_2)^2 + (\omega_1 \omega_4)^2 (\Delta q_3 - \Delta q_2)^2 + (\omega_2 \omega_3)^2 (\Delta q_4 - \Delta q_1)^2 + (\omega_2 \omega_4)^2 (\Delta q_3 - \Delta q_1)^2 + (\omega_3 \omega_4)^2 (\Delta q_2 - \Delta q_1)^2}{(\omega_1 \omega_2 \omega_3)^2 + (\omega_1 \omega_2 \omega_4)^2 + (\omega_1 \omega_3 \omega_4)^2 + (\omega_2 \omega_3 \omega_4)^2} \tag{19}$$

where $\omega$ is the Weber fractions and $\Delta$S is expressed in the unit of just noticeable difference (JND), with 1 JND corresponding to the detection threshold. Spectral sensitivity is the inverse of $I_t$ at $S = 1$ JND. We assume that receptor noise is independent of receptor type and inversely proportional to the relative frequency of the receptor types within the retina by (*Vorobyev and Osorio, 1998*), which allow us to specify the Weber fraction as:

$$\omega = \frac{v_i}{\sqrt{\eta_i}}, \tag{20}$$

where $v$ is the noise-to-signal ratio and $\eta$ is the number of receptors per receptive field. *Equation 17* accounts for the improvement in signal robustness given by spatial summation. We set the relative cone abundance to 1:2:2:4 for SWS1:SWS2:MWS:LWS, and we fix the Weber fraction to 0.05 in the LWS cone mechanism.

## Acknowledgements

We thank Susan Shen, Connie Myers, and other members of the Corbo lab for their support and constructive comments on the manuscript. Gaya Amarasinghe and Daisy Leung provided valuable assistance with enzyme expression. We thank Alex Moise for his assistance and input on the project. We thank Karen Sweazea, Emil Bautista, Andrew B Johnson, C Jonathan Schmitt for assistance providing tissue samples used in our comparative analysis.

## Additional information

### Funding

| Funder | Grant reference number | Author |
| --- | --- | --- |
| National Science Foundation | 1202776 | Matthew B Toomey |
| McDonnell Center for Cellular And Molecular Neurobiology | | Matthew B Toomey |
| National Institutes of Health | T32EY013360 | Matthew B Toomey |
| Vetenskapsrådet | 637-2013-388 | Olle Lind |
| National Institutes of Health | R01EY01157 | Rikard Frederiksen M Carter Cornwall |
| National Institutes of Health | RO1HL049879 | Robert W Curley Ken M Riedle Steven J Schwartz Earl H Harrison |
| National Institutes of Health | P30 CA016058 | Robert W Curley Ken M Riedle Steven J Schwartz Earl H Harrison |
| Air Force Office of Scientific Research | FA8655-12-1-2112 | David Wilby Nicholas W Roberts |
| Engineering and Physical Sciences Research Council | EP/E501214/1 | David Wilby Nicholas W Roberts |
| National Science Foundation | DEB-1146491 | Christopher C Witt |
| Human Frontier Science Program | RGP0017/2011 | Almut Kelber Joseph C Corbo Nicholas W Roberts |
| National Science Foundation | IOS-0910357 | Kevin J McGraw |
| Vetenskapsrådet | 2012-2212 | Almut Kelber |
| Knut och Alice Wallenbergs Stiftelse | | Almut Kelber |
| National Institutes of Health | RO1EY018826 | Joseph C Corbo |
| National Institutes of Health | RO1EY024958 | Joseph C Corbo |
| National Institutes of Health | P30EY002687 | Joseph C Corbo |
| Research to Prevent Blindness | | Joseph C Corbo |

The funders had no role in study design, data collection and interpretation, or the decision to submit the work for publication.

### Author contributions

MBT, Conception and design, Acquisition of data, Analysis and interpretation of data, Drafting or revising the article; OL, RF, RWC, KMR, CCW, MCC, Acquisition of data, Analysis and interpretation of data, Drafting or revising the article; DW, NWR, Drafting or revising the article, Contributed unpublished essential data or reagents; SJS, EHH, Analysis and interpretation of data, Contributed unpublished essential data or reagents; MV, AK, Analysis and interpretation of data, Drafting or revising the article; KJM, Conception and design, Contributed unpublished essential data or reagents; JCC, Conception and design, Analysis and interpretation of data, Drafting or revising the article

### Author ORCIDs

Matthew B Toomey, http://orcid.org/0000-0001-9184-197X
David Wilby, http://orcid.org/0000-0002-6553-8739
Christopher C Witt, http://orcid.org/0000-0003-2781-1543
Nicholas W Roberts, http://orcid.org/0000-0002-4540-6683

Almut Kelber, http://orcid.org/0000-0003-3937-2808
Joseph C Corbo, http://orcid.org/0000-0002-9323-7140

### Ethics

Animal experimentation: This study was conducted following the recommendations of Guide for the Care and Use of Laboratory Animals of the National Institutes of Health. All of the animals were handled according to approved protocols of Animal Studies Committee at Washington University in St. Louis (ASC protocol no. 20140072) and University of New Mexico Institutional Animal Care and Use Committee (protocol no. 08UNM033-TR-100117).

## Additional files

**Supplementary files**

• Supplementary file 1. The species included in our phylogenetic comparison of retina apocarotenoid composition. The tuning of the SWS1 opsin is inferred from the amino acid at position 90 of the second transmembrane helix (*Ödeen and Håstad, 2013*; *2009*). The amino acid sequence was either derived from previously published studies or was determined by sequencing of genomic DNA in the current study as indicated.

• Supplementary file 2. The species and visual system parameters used to model avian color discrimination.

• Supplementary file 3. The PCR primers used in studies of enzyme function and expression. (a) PCR primers used to clone in situ hybridization templates. (b) Primers used for qPCR quantification of apocarotenoid-metabolizing enzyme transcript expression in developing chicken retinas. (c) PCR primers used to clone full-length transcripts of apocarotenoid-metabolizing enzymes for cloning into the pTre expression vector.

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
