## [Decision Letter]

Thank you for submitting your article "Complementary shifts in photoreceptor spectral tuning unlock the full adaptive potential of ultraviolet vision in birds" for consideration by *eLife*. Your article has been favorably evaluated by Daniel Osorio, Linda Carvalho, and Fred Rieke, who is a member of our Board of Reviewing Editors. The evaluation was overseen by Eve Marder as the Senior editor.

The reviewers have discussed the reviews with one another and the Reviewing Editor has drafted this decision to help you prepare a revised submission.

All three reviewers commented on the comprehensive nature of the study and the importance of the research topic. A specific area of concern was the modeling of color discrimination. Two specific issues came up in this regard: (1) whether the model of cone noise is accurate; and (2) how the conclusions of the model depend on the spectra distribution of inputs. Because the modeling is an essential piece of the paper, and indeed one of the most novel aspects of the paper, these issues need to be dealt with directly. Specifically, it would be very helpful to know how the conclusions of the model depend on the assumptions upon which it is based. Detailed comments from each reviewer follow, including elaboration of the points above. We are including the full reviews so that you can see the convergence onto the essential points above.

Reviewer #1:

This paper uses an impressive array of techniques to investigate coordinated shifts in spectral tuning of two short-wavelength sensitive photoreceptors in birds. The central finding of the paper – that shifts in the short wavelength photopigment spectral sensitivity produce coordinated shifts in the absorption properties of the oil droplet controlling spectral tuning of the neighboring photoreceptor – are well supported by the data. I am not an expert on photopigment chemistry, so will leave detailed evaluation of that part of the paper for the other reviewers. I did have some concerns about the model for the impact of these changes in color coding:

1) Cone noise model.

Investigating chromatic discrimination requires a model for noise introduced in the cones. The cone noise is modeled in the paper as a constant Weber fraction. But this appears inconsistent with direct measures of cone noise. Generally, intrinsic noise dominates total cone noise at low light levels. As light levels increase, a point is reached at which quantal fluctuations begin to contribute and in some cases dominate. This behavior is seen in Lamb and Simon (1977), Rieke and Baylor (2000), Angueyra and Rieke (2013). I suspect a noise model consistent with these measurements would alter the findings in the paper about the dependence of optimal spectral sampling on light level. If there is a solid justification for using a constant Weber fraction noise model that should be given in the context of what is known about the cones. If not, a model more in line with the cone measurements should be used.

2) Measurement of pigment mixtures in oil droplets (subsection “C-type oil droplet spectral filtering is determined by the accumulation of two apocarotenoids, galloxanthin and dihydrogalloxanthin”, third paragraph). The data showing how the measured absorption spectra were fit to obtain a mixture of carotenoids is quite important but is not shown in any detail. I think that data needs to be included, both as absorption spectra and template fits and as summary data.

Reviewer #2:

This is an exceptionally comprehensive study of how in birds filtering of the light by the SWS2 ('blue') cone, which has a single spectral type of opsin, modifies the spectral sensitivity of this photoreceptor in concert with shifts in the shorter wavelength sensitive SWS1 (Violet/UV) cone opsin spectral sensitivity. This relationship between the spectral sensitivities of the two types of photoreceptor has been known since around 2000, but has received little attention.

Similarly, the role on carotenoid metabolism in tuning the coloured oil droplets in avian cones has been studied, but to my knowledge this is far the most comprehensive investigation of carotenoid metabolism in this context.

It is therefore the link between the biochemistry and visual function that gives this work its main significance beyond the specialised fields of carotenoid metabolism and avian colour vision. One can quantitatively specify performance, here in terms of the number of discriminable colours, and hence link the trait (carotenoid metabolism) to its function (colour discrimination) in a way that is seldom possible in other domains.

Thus this is a highly novel contribution to visual evolution and ecology, which has hitherto looked at phenomena such as photoreceptor noise and spectral tuning, and the optical design of eyes, but not spectral filtering by photostable pigments (carotenoid or other). As far as I can tell the work is done impeccably, and insofar as my competence allows, I have no concerns about the findings.

I do however have two questions:

1) At no point does the paper address the question of why shifts between UVS and VS type opsin occur so often in avian evolution, or why intermediate variants do not evolve. This is, of course, not the main subject, but a natural question to ask, so it would at least be good to know that we have no idea!

2) More substantively, the model of performance evaluates the number of discriminable colours available to the different types of eye taking account of photoreceptor noise. This measure is potentially misleading:

It is a priori obvious (and certainly doesn't need the modelling done here) that reducing spectral overlap increases the volume of the colour solid so that if the Weber fractions of the different cone types are fixed (and there is no penalty for reduced photon catch) then is obviously beneficial to reduce – or ideally eliminate – spectral overlap (see Eq 12).

It is mainly when the effects of photon noise (Eq 13) are taken into account that the modelling of the kind is useful. The model estimates a 'confusion volume', and hence the number of discriminable colours under a natural illuminant. These volumes depend upon photon noise, and hence quantum catch. This raises two questions: i) what are the spectral compositions of the stimuli that are being discriminated, and ii) how do they relate to the reflectances of real or even hypothetical objects?

For example, if spectra are generated at random – say (most unrealistically) with a reflectance of either 0 or 1 at each 1-nm interval – then it is easy to see that the density of spectra falling in the central part of the colour space will greatly exceed that in the periphery potentially reducing the benefit of minimising spectral overlap.

In reality most potentially discriminable colours, perhaps 90% – 99% of the 8-million or so, will hardly ever occur, whereas most natural spectra will reside in a very small proportion of these colour spaces. It should therefore be explained how the estimates of the number of discriminable colours might be relevant to any natural task.

Therefore, although it is intuitively reasonable that photoreceptor spectral sensitivities should overlap 'by a certain amount but not too much' I do wonder if the number of colours is a misleading characterisation of the utility of the birds' colour vision.

Reviewer #3:

Toomey and colleagues present here in this study a thorough investigation of the tuning mechanisms behind avian vision in the shorter wavelength spectrum. Birds have been shown to have their shortwave-sensitive pigments (SWS1) tuned either to the violet (VS) or the ultraviolet (UVS) spectrum and this SWS1 shift is accompanied by a same direction shift in the tuning of SWS2 pigments. Colour vision in birds is achieved by the presence of 4 types of cone opsins but is also regulated by the presence of oil droplets in cone photoreceptors. In this study, the authors demonstrate that the SWS2 tuning shift is mediated by modulating the ratio of 2 carotenoids (dihydrogalloxanthin and galloxanthin) present in the oil droplet found in SWS2 cones. They elucidate the molecular structure and identity of one of these carotenoids which was previously unknown and go on to report the ratio of these two carotenoids in 21 and 24 species of VS and UVS birds, respectively, showing a correlation between the ratios, SWS2 tuning and VS/UVS pigments. They show that shorter shifted SWS2 pigments are found in UVS species and have a higher amount of dihydrogalloxanthin while VS species have more galloxanthin. They then further investigate the possible enzymatic pathway from where dietary zeaxanthin is converted to dihydrogalloxanthin and galloxanthin through a 3-step enzymatic process. They show in vitro evidence that their proposed pathway can generate these 2 carotenoids and further confirm expression of the enzymes in the avian retina during development. Finally, creating a modelling system of the avian UVS and VS visual system, Toomey and colleagues were able to investigate the role of the complementary SWS1 tuning and SWS2 oil-droplets in colour discrimination in both VS and UVS species.

I believe this is a thorough study of the tuning mechanisms used by avian species to modulate their visual system and consequently colour discrimination. The authors offer a complete hypothesis of how this process works and what are the direct effect on the visual system of these species.

[Editors' note: further revisions were requested prior to acceptance, as described below.]

Thank you for resubmitting your work entitled "Complementary shifts in photoreceptor spectral tuning unlock the full adaptive potential of ultraviolet vision in birds" for further consideration at *eLife*. Your revised article has been favorably evaluated by Eve Marder as the Senior editor and Fred Rieke as the Reviewing editor.

The manuscript has been improved but there are a few remaining issues that need to be addressed before acceptance, as outlined below:

1) All figure supplements should be referred to in the main text. One place this comes up in particular is in describing the spectral fits in Figure 3, where the figure supplement really helps visualize the fitting procedure. Some of the figure supplements would benefit from a bit more labelling or other edits as well, e.g.: Figure 2—figure supplement 4: some of text pretty small, and lines on figure are thin Figure 3—figure supplement 2: more labels, e.g. 'template' spectra

2) Subsection “Complementary SWS1 opsin tuning and C-type oil droplet spectral filtering facilitate color discrimination”: Reviewer #2 raised a concern about the actual color space relevant for vision, and whether the analysis of the number of discriminable colors was relevant. Other readers may share this concern. Hence some of the response to this concern could make its way into the paper, and the introduction to this section seems like a good place. This could entail describing in a few sentences the 'ideal' analysis, based on known input spectra for each species natural environment, and why this is not feasible currently.

3) In the first paragraph of the subsection “C-type oil droplet spectral filtering is determined by the accumulation of two apocarotenoids, galloxanthin and dihydrogalloxanthin”: define phylogenetic inertia.

4) In the second paragraph of the subsection “A proposed enzymatic pathway for apocarotenoid metabolism in the SWS2 cone”: LWS is a bit distracting here – perhaps mention that part of your hypothesis is that these observations about expression also hold for SWS cones.

---

## [Author Response]

*All three reviewers commented on the comprehensive nature of the study and the importance of the research topic. A specific area of concern was the modeling of color discrimination. Two specific issues came up in this regard: (1) whether the model of cone noise is accurate; and (2) how the conclusions of the model depend on the spectra distribution of inputs. Because the modeling is an essential piece of the paper, and indeed one of the most novel aspects of the paper, these issues need to be dealt with directly. Specifically, it would be very helpful to know how the conclusions of the model depend on the assumptions upon which it is based. Detailed comments from each reviewer follow, including elaboration of the points above. We are including the full reviews so that you can see the convergence onto the essential points above.*

*Reviewer #1:*

*This paper uses an impressive array of techniques to investigate coordinated shifts in spectral tuning of two short-wavelength sensitive photoreceptors in birds. The central finding of the paper – that shifts in the short wavelength photopigment spectral sensitivity produce coordinated shifts in the absorption properties of the oil droplet controlling spectral tuning of the neighboring photoreceptor – are well supported by the data. I am not an expert on photopigment chemistry, so will leave detailed evaluation of that part of the paper for the other reviewers. I did have some concerns about the model for the impact of these changes in color coding:*

*1) Cone noise model.*

Investigating chromatic discrimination requires a model for noise introduced in the cones. The cone noise is modeled in the paper as a constant Weber fraction. But this appears inconsistent with direct measures of cone noise. Generally, intrinsic noise dominates total cone noise at low light levels. As light levels increase, a point is reached at which quantal fluctuations begin to contribute and in some cases dominate. This behavior is seen in Lamb and Simon (1977), Rieke and Baylor (2000), Angueyra and Rieke (2013). I suspect a noise model consistent with these measurements would alter the findings in the paper about the dependence of optimal spectral sampling on light level. If there is a solid justification for using a constant Weber fraction noise model that should be given in the context of what is known about the cones. If not, a model more in line with the cone measurements should be used.

The bird species we model in this study are primarily diurnal (i.e., active from sunrise to sunset). In diurnal chickens and budgerigars, behavioral measures of spectral sensitivity thresholds can be explained by a constant Weber fraction at light levels greater than or equal to sunset conditions (approx. 10 cd m^-2^ for chicken and 1 cd m^-2^ for budgerigar, Lind and Kelber 2009, J. Exp Biol. 212:3693 Olsson et al. 2015, J. Exp Biol. 218:184). Therefore, for the major part of the day when the birds are active, cone noise is likely to display Weber behavior.

However, to investigate a wider range of potential light conditions we have expanded our modeling to include three light levels:

A) Bright light: we assume that intrinsic noise (which scales with intensity in a Weber-like fashion) and photon shot noise are important. In chickens, behavioral thresholds consistent with Weber-like noise are seen at light levels ≥ 10 cd m^-2^. Photon shot noise likely limits discrimination of very dim colors even under bright light conditions. Here we have modeled discrimination with a Weber fraction of 0.1 in the LWS cone, based upon behavioral data on incremental spectral sensitivity in red-billed leiothrix (Maier, 1992, J. Comp. Physiol. A. 170:709) and budgerigars (Lind et al., 2014, J. Comp. Physiol. A. 200:197).

B) Dim light: we assume photon-shot noise dominates and depends upon light intensity in a linear fashion. In chickens, behavioral thresholds consistent with photo-shot noise are observed at light levels of 0.1-10 cd m^-2^ (Olsson et al. 2015, J. Exp Biol. 218:184).

C) Very dim light: we assume dark noise dominates and is invariant with respect to light intensity. In chickens, behavioral thresholds consistent with dark noise are observed at light levels of 0.025-0.1 cd m^-2^ (Olsson et al. 2015, J. Exp Biol. 218:184). Estimates of dark noise in avian cone photoreceptors are not currently available, and we have therefore chosen an arbitrary level for these calculation. The specific value of the noise parameter does not change our prediction of the relative performance of the various cone/oil droplet configuration. However, the estimates of absolute number of discriminable colors cannot be validated without direct measurement of dark noise.

These three conditions yield different predictions as to the positioning of the C-type droplet spectral filtering that will maximize the number of discriminable colors. In the bright condition, where we assume Weber-like scaling of the noise, the model predictions are consistent with the observed droplet filtering in VS and UVS species. In both dim and very dim conditions the model predictions are substantially short-wavelength shifted relative to the observed values. A short-wavelength shift in C-type droplet filtering increases spectral overlap between the SWS2 and SWS1 cones, but also increases the quantum catch of the receptors. As we have modeled the dim and very dim conditions, the benefit of increased quantum catch outweighs the cost of increased spectral overlap because the discrimination thresholds are directly dependent on intensity. Together these results indicate that in the diurnal species we examine, spectral filtering of the C-type oil droplets is fine-tuned for color discrimination in bright light conditions where discrimination thresholds display Weber-like behavior.

However, this may not be the case for all bird species. Nocturnal birds such as owls are reported to have pale, carotenoid-deficient oil droplets with much reduced spectral filtering (Bowmaker and Martin 1978, Vision Res. 18:1125). Across a broader taxonomic scale, oil droplets are often depigmented or completely lost in nocturnal avian taxa (Walls 1942, The vertebrate eye and its adaptive radiation). This suggests that oil droplet spectral filtering is not adaptive for nocturnal vision.

2) Measurement of pigment mixtures in oil droplets (subsection “C-type oil droplet spectral filtering is determined by the accumulation of two apocarotenoids, galloxanthin and dihydrogalloxanthin”, third paragraph). The data showing how the measured absorption spectra were fit to obtain a mixture of carotenoids is quite important but is not shown in any detail. I think that data needs to be included, both as absorption spectra and template fits and as summary data.

We have added detailed plots of the absorbance spectra and fitted curves for each individual droplet measured from each of the species in Figure 3—figure supplement 2.

*Reviewer #2:*

*(…] Thus this is a highly novel contribution to visual evolution and ecology, which has hitherto looked at phenomena such as photoreceptor noise and spectral tuning, and the optical design of eyes, but not spectral filtering by photostable pigments (carotenoid or other). As far as I can tell the work is done impeccably, and insofar as my competence allows, I have no concerns about the findings.*

*I do however have two questions:*

1) At no point does the paper address the question of why shifts between UVS and VS type opsin occur so often in avian evolution, or why intermediate variants do not evolve. This is, of course, not the main subject, but a natural question to ask, so it would at least be good to know that we have no idea!

The molecular mechanism of SWS1 opsin spectral tuning offers an explanation for why these transitions occur so frequently and why intermediate tunings are not typically observed. The shift of the λ_max_ of SWS1 opsin from VS to UVS can be affected by as little as a single amino acid change in the opsin (S90C). In fact, this amino acid substitution can be brought about by a single nucleotide change at the DNA level. For this reason, the VS to UVS shift is easily ‘accessible’ in an evolutionary sense. The S90C substitution is predicted to cause deprotonation of the chromophore Schiff-base linkage in the opsin (Carvalho et al. 2007, Mol. Bio. Evol. 24:183, Altun et al. 2011, ACS Chem. Biol. 6:775). This deprotonation results in a large blue-shift in the peak spectral sensitivity that dwarfs more subtle shifts brought about by other changes in the opsin. Therefore, the evolution of intermediate tunings may be constrained by the photochemistry of the visual pigment.

At the level of ultimate causation, currently relatively little is understood about the selective pressures that might be driving the VS to UVS shift. This is, of course, a very interesting question. However, much of this discussion is speculative and to include it here would require an extensive Introduction and Discussion that is beyond the scope of the paper. Nevertheless, we have edited lines (Introduction, second paragraph) to acknowledge this question.

*2) More substantively, the model of performance evaluates the number of discriminable colours available to the different types of eye taking account of photoreceptor noise. This measure is potentially misleading:*

*It is a priori obvious (and certainly doesn't need the modelling done here) that reducing spectral overlap increases the volume of the colour solid so that if the Weber fractions of the different cone types are fixed (and there is no penalty for reduced photon catch) then is obviously beneficial to reduce – or ideally eliminate – spectral overlap (see Eq 12).*

*It is mainly when the effects of photon noise (Eq 13) are taken into account that the modelling of the kind is useful. The model estimates a 'confusion volume', and hence the number of discriminable colours under a natural illuminant. These volumes depend upon photon noise, and hence quantum catch. This raises two questions: i) what are the spectral compositions of the stimuli that are being discriminated, and ii) how do they relate to the reflectances of real or even hypothetical objects?*

*For example, if spectra are generated at random – say (most unrealistically) with a reflectance of either 0 or 1 at each 1-nm interval – then it is easy to see that the density of spectra falling in the central part of the colour space will greatly exceed that in the periphery potentially reducing the benefit of minimising spectral overlap.*

*In reality most potentially discriminable colours, perhaps 90% – 99% of the 8-million or so, will hardly ever occur, whereas most natural spectra will reside in a very small proportion of these colour spaces. It should therefore be explained how the estimates of the number of discriminable colours might be relevant to any natural task.*

*Therefore, although it is intuitively reasonable that photoreceptor spectral sensitivities should overlap 'by a certain amount but not too much' I do wonder if the number of colours is a misleading characterisation of the utility of the birds' colour vision.*

This is a good point and a challenge we have discussed extensively. We agree that it is unlikely that the spectral world of any given bird species completely fills color space. However, we have a very limited understanding of the spectra that are available and important for specific bird species. Restricting our analyses to specific regions of color space would largely be arbitrary and could also be misleading. Therefore, we chose to limit our assumptions and model the entirety of color space.

Our calculations of the spectral sensitivity function indicate that C-type spectral shifts have their most pronounced effect in the 'blue' region (400-500 nm). If we were to restrict our analysis to this region, our results would possibly be somewhat clearer, but less general.

[Editors' note: further revisions were requested prior to acceptance, as described below.]

*The manuscript has been improved but there are a few remaining issues that need to be addressed before acceptance, as outlined below:*

*1) All figure supplements should be referred to in the main text. One place this comes up in particular is in describing the spectral fits in Figure 3, where the figure supplement really helps visualize the fitting procedure. Some of the figure supplements would benefit from a bit more labelling or other edits as well, e.g.:*

Figure 2—figure supplement 4: some of text pretty small, and lines on figure are thin Figure 3—figure supplement 2: more labels, e.g. 'template' spectra

We have added references to Figure 3—figure supplement 2 and revised the third paragraph of the subsection “C-type oil droplet spectral filtering is determined by the accumulation of two apocarotenoids, galloxanthin and dihydrogalloxanthin” to clarify the spectral fitting estimates of carotenoid composition.

We have added descriptive labels to each subsection of Figure 3—figure supplement 2 to clarify the material presented.

We have revised Figure 2—figure supplement 4 and increased lines weights and font size.

2) Subsection “Complementary SWS1 opsin tuning and C-type oil droplet spectral filtering facilitate color discrimination”: Reviewer #2 raised a concern about the actual color space relevant for vision, and whether the analysis of the number of discriminable colors was relevant. Other readers may share this concern. Hence some of the response to this concern could make its way into the paper, and the introduction to this section seems like a good place. This could entail describing in a few sentences the 'ideal' analysis, based on known input spectra for each species natural environment, and why this is not feasible currently.

We have added a discussion of the relevance of the discriminable color analysis (subsection “Complementary SWS1 opsin tuning and C-type oil droplet spectral filtering facilitate color discrimination”, first paragraph). We acknowledge that ideally we would limit the analysis to the colors that are ecologically relevant for a given species. However, to date, such a “chromatic ecosystem” has not been defined for any species and we chose to analyze all of color space in order to limit the assumptions we make.

3) In the first paragraph of the subsection “C-type oil droplet spectral filtering is determined by the accumulation of two apocarotenoids, galloxanthin and dihydrogalloxanthin”: define phylogenetic inertia.

We have clarified the meaning of phylogenetic inertia in the first paragraph of the subsection “C-type oil droplet spectral filtering is determined by the accumulation of two apocarotenoids, galloxanthin and dihydrogalloxanthin”.

4) In the second paragraph of the subsection “A proposed enzymatic pathway for apocarotenoid metabolism in the SWS2 cone”: LWS is a bit distracting here – perhaps mention that part of your hypothesis is that these observations about expression also hold for SWS cones.

We have revised this statement to clarify the logic of our hypothesis (subsection “A proposed enzymatic pathway for apocarotenoid metabolism in the SWS2 cone”, second paragraph).